# LA–ICP–MS U–Pb Dating, Elemental Mapping and In Situ Trace Element Analyses of Cassiterites from the Gejiu Tin Polymetallic Deposit, SW China: Constraints on the Timing of Mineralization and Precipitation Environment

**Xiaohu He** [1] ⓘ, **Congfa Bao** [2], **Yongyan Lu** [2], **Nicole Leonard** [3], **Zheng Liu** [1] **and Shucheng Tan** [1,*]ⓘ

[1] School of Earth Scieces, Yunnan University, Kunming 650500, China; xhhe@ynu.edu.cn (X.H.); zhengliu@lzu.edu.cn (Z.L.)
[2] Yunnan Planning and Design Institute for Mineral and Land Resources, Kunming 650216, China; baocf2022@163.com (C.B.); lu1103210@163.com (Y.L.)
[3] School of Earth and Environmental Sciences, The University of Queensland, Brisbane, QLD 4072, Australia; n.leonard@uq.edu.au
* Correspondence: tansc001@126.com; Tel.: +86-0871-6503-3733

**Abstract:** As a major constituent in magmatic–hydrothermal ore deposits, cassiterites, with moderate amounts of U and low Pb, can be dated with U–Pb geochronology. The tetragonal lattice structure makes cassiterites capable of incorporating dozens of elements within its crystal lattice (e.g., Fe, Ti, W, Zr, Hf, Ta, Nb, Mn, Sc, V, and Sb). Variations of these elements record information of potential elemental substitution mechanisms and precipitation environments of cassiterites. In this study, we collected cassiterite grains from four different ore styles of the Gejiu tin polymetallic deposit to perform LA–ICP–MS U–Pb dating, multiple element mapping, and in situ trace element analysis on these cassiterites. Systematic U–Pb dating yielded Tera–Wasserburg lower intercepted ages at around 85 Ma, coinciding with zircon U–Pb ages of regional Late Yanshanian granitoids, within their respective analytical uncertainties. Such age coincidence, combined with the spatial association, suggests that tin mineralization may be genetically related to the Late Cretaceous granitic magmatism. Multielemental mapping shows that the distribution of Nb, Ta, and Ti in the cassiterite grains correlates well with the regular oscillatory zoning patterns in cathodoluminescence (CL) images. The relatively high Sb, Fe, W, Ga, and U concentrations control the dark luminescing domains in these cassiterite grains. The systematic variations in chemical compositions suggest that trace elements such as Sc, V, Fe, and Ga incorporate in cassiterites via coupled substitutions of $Sc^{3+} + V^{5+} \leftrightarrow 2 (Sn, Ti)^{4+}$, $Fe^{3+} + Ga^{5+} \leftrightarrow 2 (Sn, Ti)^{4+}$ and $Fe^{3+} + OH^- \leftrightarrow Sn^{4+} + O^{2-}$ or $Fe^{3+} + H^+ \leftrightarrow Sn^{4+}$. The covariation of redox sensitive elements such as W, U, Fe, and Sb indicates that the "tin-granite" type of cassiterites were formed under an oxidized state whereas cassiterites from skarn, massive sulfide, and oxidized ore styles were precipitated in a reducing environment.

**Keywords:** cassiterite; LA–ICP–MS mapping; U–Pb dating; trace element; precipitation environment; Gejiu

## 1. Introduction

Cassiterite ($SnO_2$) is the most important ore mineral for tin deposits and has a tetragonal lattice structure similar to that of the rutile group ($M^{4+}O_2$) [1–4]. Because cassiterite is resistant to chemical and physical alteration and weathering, trace elements, such as Fe, Ti, W, Zr, Hf, Ta, Nb, Mn, Sc, V, and Sb in cassiterite could be used to study mineralization processes and precipitation environments of tin mineralization [2–10]. Furthermore, the presence of U and Pb makes it possible to date cassiterite with U–Pb

geochronology [1,2,4,7–9,11–15]. Thus, cassiterite can be used to investigate mineralization timing, precipitation environment, and mechanisms for magmatic–hydrothermal ore deposits or important tin belts worldwide [3,4,16,17].

The world-class Gejiu tin polymetallic deposit in SW China has large metal reserves with an endowment estimated at ~10 million metric tonnes of tin, copper, lead, and zinc, containing approximately 300 Mt of tin and copper ore averaging 1 wt% Sn and 2 wt% Cu, respectively, and 400 Mt of lead–zinc ore with 7 wt% Pb + Zn [3,18,19]. Many attempts at dating mineralization timing have been conducted on hydrothermal minerals, including K–Ar/$^{40}$Ar–$^{39}$Ar dating on quartz/cassiterite [20], Pb–Pb dating on sulfide [20], Re–Os dating of molybdenite [21], and U–Pb dating of cassiterite [3,8,10,22]. However, there is still a lack of systematic cassiterite U–Pb dating of different ore styles. To trace the origins of ore-forming elements, many studies have focused on stable isotope compositions [23,24], but few studies have examined the composition of fluid inclusions to interpret the precipitation environments of the Gejiu deposit [10,22,24,25]. Thus, precipitation environments of cassiterite in hydrothermal fluids are poorly understood. Here, we study cassiterite grains from the Gejiu tin polymetallic deposit with U–Pb geochronology, elemental mapping, and in situ trace element geochemistry. Results allow us to discuss (1) the timing of Sn mineralization, (2) trace element concentration variations and elemental zoning in cassiterite, and (3) elemental substitution mechanisms and precipitation environments of cassiterites. These data could provide further insights into ore genesis in the world-class Gejiu tin district whilst highlighting the potential for using cassiterites as a monitor of hydrothermal processes and for understanding the timing and evolution of Sn mineralization events.

## 2. Regional Geological Setting and Deposit Geology

The Gejiu deposit in SW China is located along the boundary between the Tethys and Pacific tectonic domains and adjacent to the Ailao Shan–Red River (ASRR) shear zone (see Figure 1a). It is also adjacent to the Youjiang basin, which is bounded by the Mile–Shizong fault to the northwest, the Nandan–Duan fault to the northeast, the Pingxiang–Nanning fault to the southeast, and the NW-trending Honghe fault to the southwest (see Figure 1b, [26–28]). In the Gejiu mining district, almost all tin deposits are distributed to the east of the Gejiu fault, with numerous plutons but few deposits in the west. It can be divided into five east–west trending zones by major faults named Malage, Songshujiao, Gaosong, Laochang, and Kafang, from north to south into five ore fields. Extensive magmatic activity led to the development of numerous plutons consisting of gabbros, porphyritic and biotite granites, syenites, and metabasalts (see Figure 2, [29,30]).

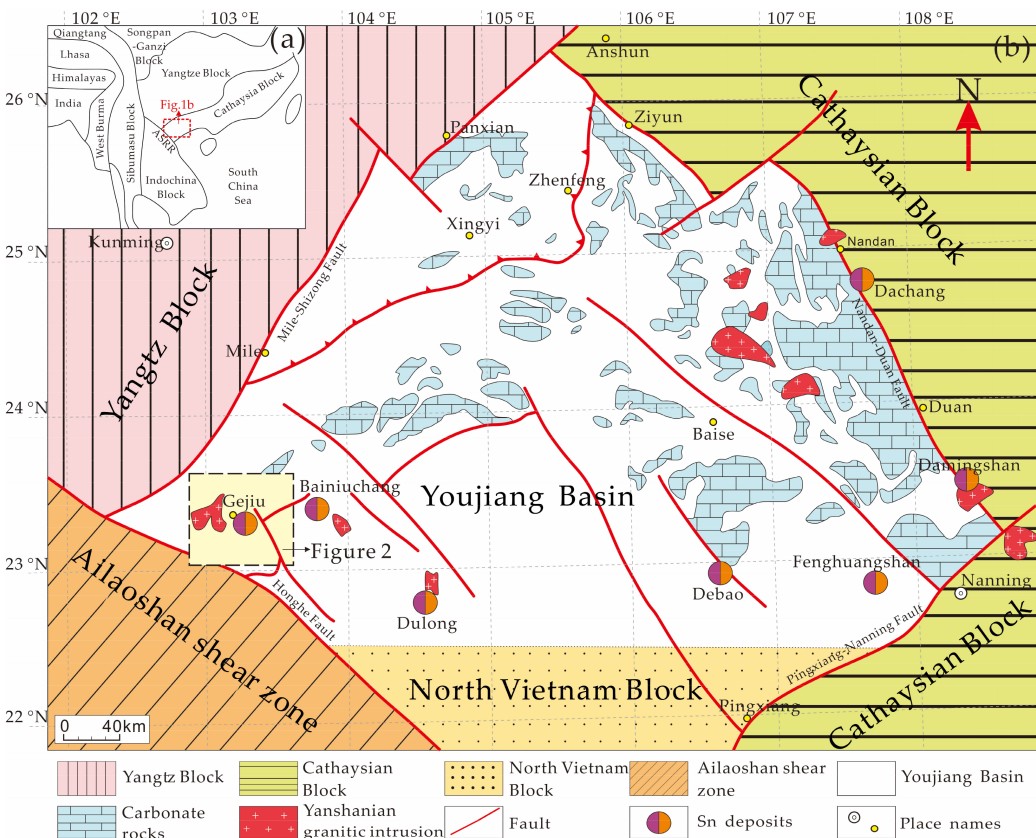

**Figure 1.** (**a**) Simplified geological map of eastern Asia, showing the major tectonic units [25,31], ASRR = Ailao Shan–Red River shear zone. (**b**) Geological map showing the distribution of tin deposits, including Gejiu in the Youjiang Basin, SW China (modified from [27,28]).

Previous studies suggest that granitoids in the Gejiu district were emplaced between 85 and 77 Ma [32,33], which is consistent with the Re–Os and $^{40}$Ar–$^{39}$Ar ages (86 Ma ~ 77 Ma) of various ores from this deposit [10,30]. Excepting the absence of Late Triassic to Cretaceous strata due to episodic uplift, or erosion associated with Indosinian and Yanshanian tectonic events, Phanerozoic sedimentary stratum are well preserved in this district [8,18]. The strata largely consist of Middle Triassic Gejiu Formation carbonate sequences and the Falang Formation, which contains fine-grained clastic and carbonate sediments with intercalated mafic–felsic lavas (Figure 2b). Owing to extensive hydrothermal alteration, there are six ore styles, mainly composed of massive sulfide, oxidized ores, tin-granite, skarn, greisen, and veined tourmaline. Among them, skarn ores are the most widely distributed across the ore district and the most economically important [3]. Previous studies suggested that, except for the oxidized Sn ores, the other five types of Sn ores are considered to be related to the Yanshanian granitic magmatic activity [19,23,34]. However, several other studies suggested that the origin of the oxidized Sn ores is also closely related to the Gejiu granitic magmatic activity [25,29,30,33,35].

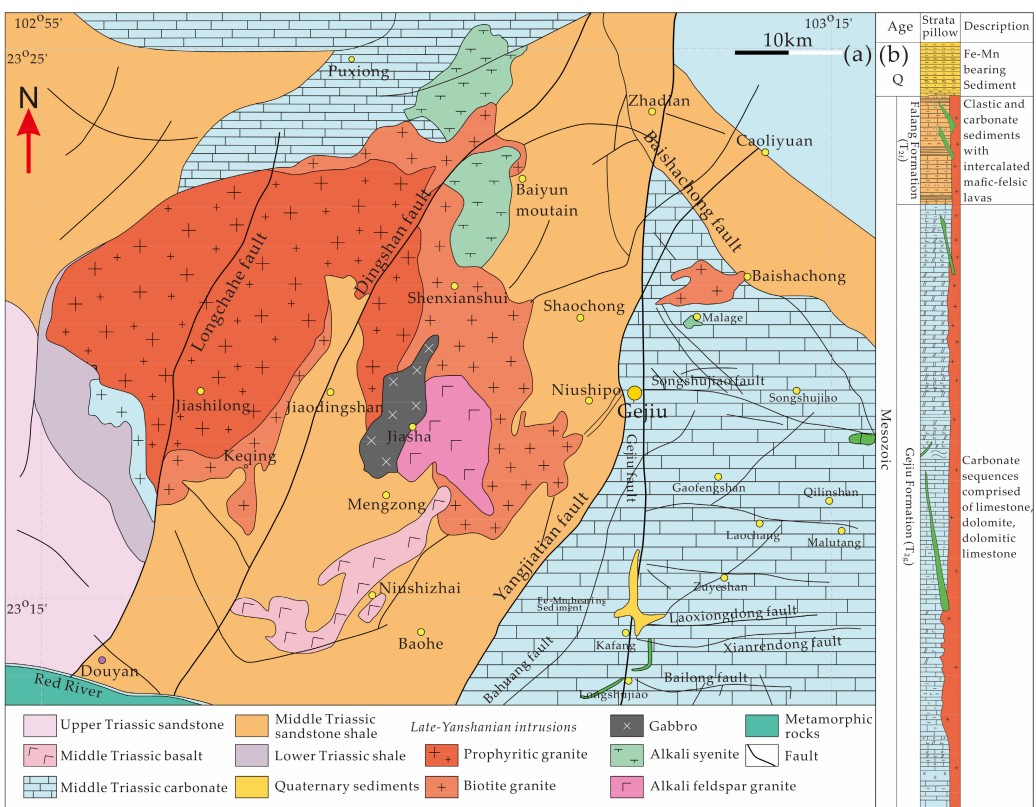

**Figure 2.** Regional geological map of the Gejiu ore district (**a**) and its strata (**b**) (modified from [3]).

## 3. Sample Description

We collected six cassiterite samples representing four different ore styles in the Gejiu Sn–polymetallic deposit. They include tin-granite (KF–03), skarn (KF–05), massive sulfide (LC–01, LC–02), and oxidized ores (SK–04, SK–05). Representative photographs of tin ores are shown in Figure 3. Cassiterite-bearing granite (hosting tin-granite) is not common in the ore district and considered as altered rock in a previous study [10]. Cassiterites hosted by tin-granite are dispersed in the granite, and their grains are always intergrown with fluorite and accompanied by tourmaline (Figure 3a). The skarn ores are mainly pyroxene- and garnet-rich with associated tremolite, actinolite, epidote, chlorite, fluorite, and tourmaline, with major ore minerals such as cassiterite, pyrrhotite, and chalcopyrite (Figure 3b). Massive sulfide ores comprise sulfides such as chalcopyrite, pyrrhotite, fluorite, azurite, scheelite, and cassiterite (Figure 3c). The oxidized ores are considered as loose earthy in this study; in previous studies, they may be named semioxidized stratiform or loose earthy ores in Triassic carbonate layers distal to granite [22,28]. Usually, they were thought to be products of the oxidation of primary sulfide ores and are dominated by hematite, limonite, pyrrhotite, and minor pyrite (Figure 3d). Most of them are hosted in Triassic carbonate layers and close to the surface and associated with small-scale faults [22].

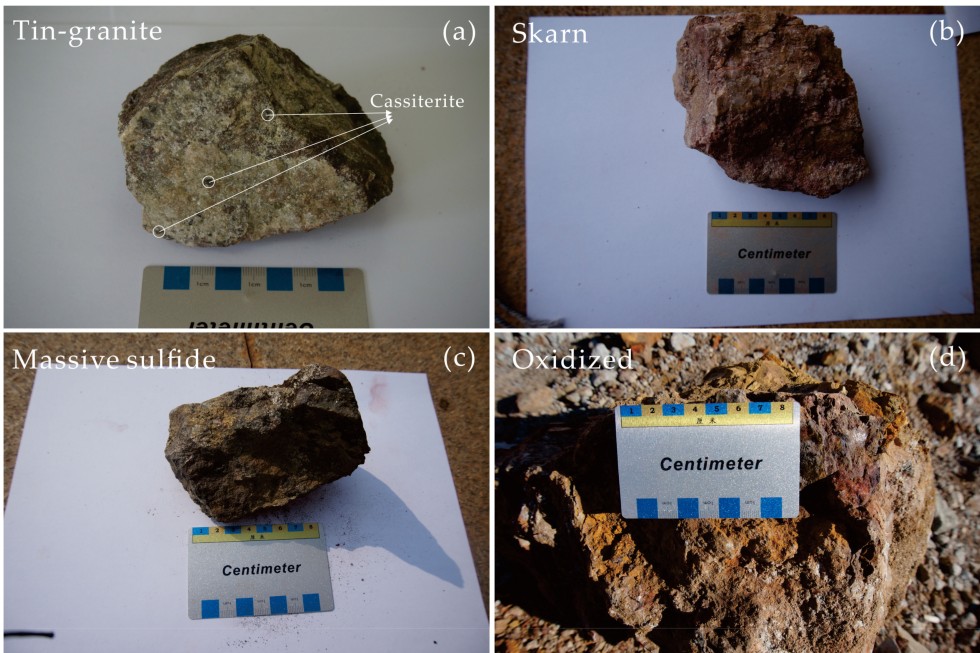

**Figure 3.** Representative photographs of tin ores from the Gejiu tin polymetallic deposit. (**a**) Tin-granite ore, also called cassiterite-bearing granite, which is only well developed in the interior of certain granite cupolas; (**b**) skarn ore is widely distributed in the contact zone of granite and carbonate, and contains garnet, pyroxene, epidote, and fluorite; (**c**) massive sulfide ore contains sulfide minerals; (**d**) red-brown oxidized ore is hosted in carbonate layers distal to intrusion and reveals a colloidal or honeycomb structure.

## 4. Analytical Methods

### 4.1. Cathodoluminescence Imaging

All samples were crushed and processed through conventional mineral separation processes. Cassiterite grains were handpicked under a binocular microscope and mounted in epoxy resin. To document internal structures and select potential targets for U–Pb analysis, cathodoluminescence (CL) images (Figure 4) were obtained using a scanning electron microscope. Backscattered electron (BSE) and cathodoluminescence (CL) images were taken using a Zeiss Supra 55 field emission scanning electron microscope (SEM, JSM–IT100, Japan Electron Optics Laboratory Co., Ltd., Tokyo, Japan) coupled with a cathodoluminescence detector at Wuhan Sample Solution Analytical Technology Co., Ltd. (Wuhan, China). Operating conditions included an accelerating potential of 10 kV, a temperature of 20 °C, and an image acquisition time of 35 s/sheet. To compare the CL images of cassiterite grains, specific analytical conditions were adopted during the experiment such that the instrument automatically adjusted the parameters to obtain an image with the highest resolution after the scale was changed.

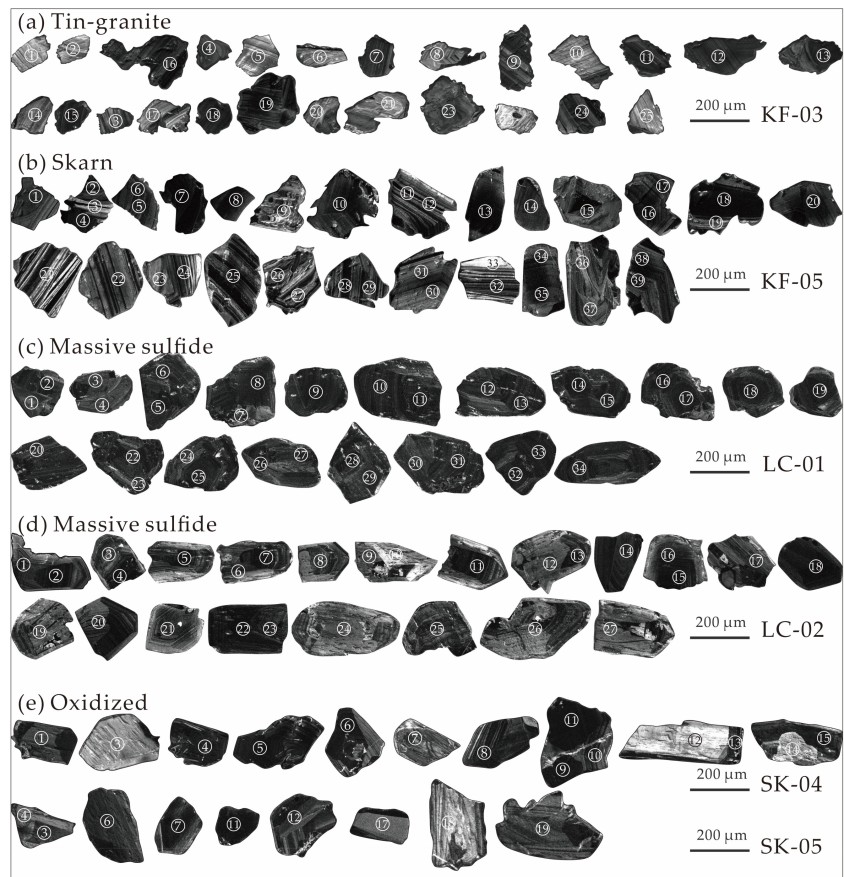

**Figure 4.** Representative cathodoluminescence images of the cassiterite samples from the Gejiu tin polymetallic deposit, showing internal textures of cassiterite grains. (**a**,**b**) Cassiterite grains from the tin–granite ore style display a regular oscillatory zoning; (**c**,**d**) Cassiterite grains from the massive sulfide ore style display an inner core with a regrowth rim; (**e**) Cassiterite grains from the oxidized ore style display an inner core and oscillatory zoning.

*4.2. Cassiterite LA–ICP–MS U–Pb Dating*

Cassiterite LA–ICP–MS U–Pb dating was performed at the Radiogenic Isotope Facility, The University of Queensland, Saint Lucia, QLD (Brisbane, Australia), using a Thermo iCAP RQ quadruple ICP–MS equipped with an ASI RESOlution SE 193 ArF nm excimer laser system. The laser was run with a 30 micron diameter round spot at 10 Hz, with a measured instrument laser-fluence (laser pulse energy per unit area) of 2.9 J/cm². For each spot, 10 s of blank was collected, followed by 20 s of ablation, and 5 s of washing. Helium gas carrying the ablated sample aerosol was mixed with argon (carrier gas) and nitrogen (additional diatomic gas) to enhance sensitivity to the ICP–MS instrument [1]. NIST–612 glass standard was employed for instrument tuning, achieving typical instrument sensitivity of about 40,000–50,000 cps/ppm $^{238}$U with 100 μm laser spot size. A working standard (cassiterite AY–4, collected from the Anyuan tin deposit of the Furong district, South China, which has been well calibrated using ID–TIMS with a U–Pb age of 158.2 ± 0.4 Ma, 14) was used to correct the $^{206}$Pb/$^{238}$U ratio. Operating conditions and analytical procedures were similar to those described by Nuriel et al. [36], Luo et al. [37], and Yang et al. [38]. Tera–Wasserburg concordia diagrams were processed using Isoplot 3.0 [39]. Data uncertainties of isotopic ratios are reported at 2σ level [1].

*4.3. In Situ Trace Elemental Analysis and Mapping*

In situ trace element analysis for cassiterite grains was performed at Wuhan Sample Solution Analytical Technology Co., Ltd., using an Agilent 7900 ICP–MS equipped with a GeolasHd 193 nm laser ablation system. Detailed operating conditions and the ICP–MS

instrument parameters are the same as those reported by Liu et al. [40]. Cassiterite grains were analyzed using a laser energy density of 3.46 J/cm$^2$, a spot size of 60 μm, and a laser pulse rate of 5 Hz. Multiple reference materials (NIST610, BCR–2G, BHVO–2G, and BIR–1G) were used as the bracketing external standard, $^{118}$Sn was used as the internal standard, assuming stoichiometric SnO$_2$ for quantification purposes. The offline selection, integration of the background and analytical signals, time-drift correction, and quantitative calibration were processed using ICPMS DataCal 10.1 (China University of Geosciences, Wuhan, China) [41].

We used an ASI RESOlution 193 nm excimer UV ArF laser ablation system with a dual-volume Laurin Technic ablation cell, coupled to a Thermo iCap RQ quadruple mass spectrometer to perform elemental mapping at the Radiogenic Isotope Facility of The University of Queensland, Australia. The laser system was operated with GeoStar Norris software and the mass spectrometer with Qtegra software. Ablation was performed in ultrapure He (grade 5.0, 99.999% purity) to which the Ar make-up gas and a trace amount of N$_2$ was added for efficient transport and to aid ionization. The instrument was tuned with scans on NIST SRM 612 glass. Elemental maps were built with Iolite [42] v3.71 in quantitative mode, using NIST SRM 612 glass as calibration standard and Sn concentrations as internal standard. Accuracy and precision were monitored using BHVO–2G, BCR–2G, BIR–1G, and GSD–1G glass reference materials as quality monitor standards (http://georem.mpch--mainz.gwdg.de/, accessed on 29 November 2021). Accuracy was typically better than 1–10% and precision better than 1–5% (*n* = 5 for each glass reference material). Limits of detection were at the sub-ppm level for most analyzed elements and ≤2 ppm for Na and Cr. In this study, three types of cassiterite grains were investigated following the rastering technique described in Ubide et al. [43].

## 5. Results

### 5.1. Internal Structure Variations of Cassiterites

Cassiterite samples can be classified into two groups based on their CL images (Figure 4). The first group includes cassiterites from tin-granite and skarn ores, revealing dull luminescing and regular oscillatory zoning (see Figure 4a,b). The second group mainly includes cassiterites from massive sulfide and oxidized ores, revealing dark gray cores with dull rims and fine oscillatory zoning overgrowth (see Figure 4c–e). In general, cassiterite grains from tin-granite and skarn ores are largely free of inclusions with the exception of occasional micrometer-sized inclusions of scheelite and Fe-oxide. Notably, sulfide and oxidized cassiterites reveal an inner core with a regrowth rim, which is considered a dissolution–reprecipitation texture, indicating that the sulfide and oxidized cassiterite crystals may have undergone a dissolution–reprecipitation process. Whilst most cassiterite grains are dark (weak CL response), others appear highly luminescent in CL, and crystals often display regular variations between bright and dark CL. Internal zoning in cassiterite grains indicates that variations in CL signals may be due to intragrain scale compositional variations, which is proved by the elemental variation shown by elemental mappings in the following text.

### 5.2. Cassiterite LA–ICP–MS U–Pb Ages

Based on reverberation and transmission photos and CL images, we chose domains without mineral or fluid inclusions and cracks to perform U–Pb isotope analyses for dating the cassiterite. The LA–ICP–MS U–Pb dating results are provided in Table 1 and Figure 5. Isotopic ratios show large variations with $^{238}$U/$^{206}$Pb in the range of 3.937–87.642 and $^{207}$Pb/$^{206}$Pb in the range of 0.047–0.849. Samples yielded Tera–Wasserburg concordia lower-intercept ages at around 85 Ma. The plots of tin-granite sample (KF–03, Figure 5a) yielded a Tera–Wasserburg concordia lower-intercept age of 85.5 ± 1.0 Ma (2σ, *n* = 29, MSWD = 2.7), the plots of skarn sample (KF–05, Figure 5b) yielded a Tera–Wasserburg concordia lower-intercept age of 85.90 ± 0.41 Ma (2σ, *n* = 39, MSWD = 0.92), the plots of tin-granite sample (LC–01, Figure 5c) yielded a Tera–Wasserburg concordia lower-intercept age of 85.2 ± 1.0 Ma (2σ, *n* = 43, MSWD = 3.7), respectively. The age results are consistent with

the zircon U–Pb ages of Late Yanshanian granitoids (77.4–85.0 Ma, according to Cheng and Mao [33]) in this district within a reasonable error range. These concordia ages represent the mineralization timing of the Gejiu tin polymetallic deposits.

**Table 1.** U–Pb isotopic ratios of cassiterite samples analyzed by LA–ICP–MS single spot analysis.

| Spot No. | U (ppm) | 2S | $^{238}U/^{206}Pb$ | 2S | $^{207}Pb/^{206}Pb$ | 2S |
|----------|---------|------|---------|-------|---------|-------|
| KF-03-01 | 1.46 | 0.04 | 3.937 | 0.527 | 0.849 | 0.023 |
| KF-03-02 | 7.55 | 0.14 | 25.253 | 4.655 | 0.442 | 0.043 |
| KF-03-03 | 52.6 | 0.41 | 82.988 | 1.928 | 0.058 | 0.004 |
| KF-03-04 | 42.6 | 0.51 | 83.612 | 1.818 | 0.050 | 0.003 |
| KF-03-05 | 70.4 | 0.70 | 84.175 | 1.630 | 0.055 | 0.003 |
| KF-03-06 | 1.02 | 0.03 | 69.444 | 8.198 | 0.262 | 0.075 |
| KF-03-07 | 67.7 | 0.77 | 60.864 | 1.148 | 0.283 | 0.007 |
| KF-03-08 | 54.4 | 0.73 | 85.106 | 1.811 | 0.048 | 0.003 |
| KF-03-09 | 44.8 | 0.40 | 82.034 | 1.884 | 0.048 | 0.003 |
| KF-03-10 | 10.8 | 0.41 | 77.761 | 3.144 | 0.087 | 0.011 |
| KF-03-11 | 40.7 | 0.31 | 83.612 | 2.027 | 0.047 | 0.003 |
| KF-03-12 | 20.3 | 0.56 | 62.305 | 2.446 | 0.267 | 0.020 |
| KF-03-13 | 9.69 | 0.21 | 18.553 | 3.098 | 0.531 | 0.043 |
| KF-03-14 | 24.6 | 0.41 | 60.241 | 3.992 | 0.241 | 0.032 |
| KF-03-15 | 2.91 | 0.0 | 67.751 | 4.269 | 0.163 | 0.025 |
| KF-03-16 | 5.78 | 0.11 | 32.573 | 1.592 | 0.564 | 0.027 |
| KF-03-17 | 10.9 | 0.45 | 82.508 | 2.995 | 0.068 | 0.006 |
| KF-03-18 | 50.3 | 0.58 | 58.824 | 3.806 | 0.254 | 0.024 |
| KF-03-19 | 3.50 | 0.22 | 77.160 | 4.823 | 0.143 | 0.023 |
| KF-03-20 | 26.8 | 0.81 | 68.306 | 1.820 | 0.188 | 0.011 |
| KF-03-21 | 18.7 | 0.77 | 31.546 | 5.175 | 0.433 | 0.036 |
| KF-03-22 | 1.96 | 0.048 | 40.984 | 4.367 | 0.559 | 0.062 |
| KF-03-23 | 0.56 | 0.039 | 18.182 | 6.612 | 0.430 | 0.100 |
| KF-03-24 | 20.3 | 0.26 | 54.855 | 1.414 | 0.367 | 0.013 |
| KF-03-25 | 1.61 | 0.07 | 69.444 | 6.269 | 0.214 | 0.041 |
| KF-03-26 | 42.3 | 0.50 | 75.643 | 2.060 | 0.112 | 0.010 |
| KF-03-27 | 40.0 | 0.40 | 75.131 | 4.911 | 0.106 | 0.021 |
| KF-03-28 | 2.82 | 0.51 | 63.291 | 6.409 | 0.286 | 0.061 |
| KF-03-29 | 44.5 | 1.60 | 83.963 | 1.762 | 0.051 | 0.003 |
| KF-05-01 | 25.4 | 0.66 | 6.075 | 0.351 | 0.781 | 0.011 |
| KF-05-02 | 4.15 | 0.14 | 56.497 | 4.150 | 0.289 | 0.036 |
| KF-05-03 | 15.6 | 0.13 | 73.746 | 1.849 | 0.128 | 0.010 |
| KF-05-04 | 3.02 | 0.18 | 79.051 | 4.249 | 0.074 | 0.014 |
| KF-05-05 | 21.6 | 0.31 | 80.451 | 1.812 | 0.073 | 0.004 |
| KF-05-06 | 9.57 | 0.15 | 72.254 | 3.289 | 0.136 | 0.016 |
| KF-05-07 | 13.0 | 0.43 | 83.403 | 2.365 | 0.060 | 0.005 |
| KF-05-08 | 12.8 | 0.22 | 82.372 | 2.036 | 0.060 | 0.004 |
| KF-05-09 | 19.4 | 0.20 | 64.516 | 1.915 | 0.220 | 0.013 |
| KF-05-10 | 2.57 | 0.06 | 79.365 | 2.331 | 0.074 | 0.007 |
| KF-05-11 | 11.4 | 0.19 | 79.745 | 2.480 | 0.071 | 0.007 |
| KF-05-12 | 3.99 | 0.13 | 79.808 | 1.847 | 0.066 | 0.005 |
| KF-05-13 | 11.6 | 0.11 | 82.034 | 1.952 | 0.061 | 0.004 |
| KF-05-14 | 3.50 | 0.15 | 76.278 | 4.596 | 0.117 | 0.021 |
| KF-05-15 | 15.9 | 0.26 | 82.034 | 2.355 | 0.068 | 0.007 |
| KF-05-16 | 13.7 | 0.11 | 80.128 | 1.926 | 0.061 | 0.005 |
| KF-05-17 | 24.5 | 0.40 | 57.438 | 1.551 | 0.297 | 0.013 |
| KF-05-18 | 22.6 | 0.67 | 76.511 | 2.400 | 0.111 | 0.008 |
| KF-05-19 | 25.9 | 2.00 | 81.037 | 1.773 | 0.069 | 0.004 |
| KF-05-20 | 14.5 | 0.43 | 53.533 | 2.149 | 0.330 | 0.019 |
| KF-05-21 | 43.3 | 0.98 | 77.160 | 3.929 | 0.072 | 0.013 |
| KF-05-22 | 17.6 | 0.43 | 63.654 | 2.634 | 0.243 | 0.020 |
| KF-05-23 | 38.4 | 2.00 | 82.919 | 1.925 | 0.057 | 0.004 |
| KF-05-24 | 33.6 | 0.64 | 82.988 | 2.273 | 0.051 | 0.004 |

**Table 1.** *Cont.*

| Spot No. | U (ppm) | 2S | $^{238}$U/$^{206}$Pb | 2S | $^{207}$Pb/$^{206}$Pb | 2S |
|---|---|---|---|---|---|---|
| KF-05-25 | 8.82 | 0.16 | 80.841 | 2.222 | 0.051 | 0.005 |
| KF-05-26 | 15.6 | 0.25 | 82.102 | 1.955 | 0.053 | 0.003 |
| KF-05-27 | 25.8 | 0.43 | 23.256 | 0.919 | 0.601 | 0.025 |
| KF-05-28 | 11.4 | 0.16 | 83.542 | 2.024 | 0.048 | 0.004 |
| KF-05-29 | 29.9 | 0.63 | 83.056 | 1.587 | 0.055 | 0.002 |
| KF-05-30 | 87.9 | 1.50 | 81.633 | 1.933 | 0.057 | 0.003 |
| KF-05-31 | 22.7 | 0.38 | 78.989 | 2.246 | 0.078 | 0.007 |
| KF-05-32 | 29.1 | 0.32 | 81.967 | 2.687 | 0.050 | 0.005 |
| KF-05-33 | 18.7 | 0.17 | 83.056 | 2.414 | 0.050 | 0.004 |
| KF-05-34 | 11.2 | 0.15 | 80.841 | 1.634 | 0.088 | 0.005 |
| KF-05-35 | 80.9 | 1.10 | 82.305 | 1.490 | 0.056 | 0.002 |
| KF-05-36 | 16.8 | 0.31 | 81.833 | 3.080 | 0.059 | 0.007 |
| KF-05-37 | 20.2 | 1.00 | 80.906 | 2.291 | 0.051 | 0.005 |
| KF-05-38 | 18.1 | 0.26 | 82.988 | 2.617 | 0.058 | 0.006 |
| KF-05-39 | 4.59 | 0.27 | 85.911 | 3.764 | 0.060 | 0.009 |
| LC-01-01 | 15.8 | 0.51 | 66.756 | 2.897 | 0.180 | 0.021 |
| LC-01-02 | 13.1 | 0.43 | 70.771 | 2.504 | 0.175 | 0.014 |
| LC-01-03 | 14.4 | 0.22 | 77.640 | 2.230 | 0.060 | 0.005 |
| LC-01-04 | 20.3 | 0.24 | 59.312 | 1.618 | 0.294 | 0.012 |
| LC-01-05 | 14.7 | 0.99 | 65.963 | 2.132 | 0.214 | 0.019 |
| LC-01-06 | 34.4 | 0.52 | 87.642 | 1.997 | 0.069 | 0.004 |
| LC-01-07 | 10.6 | 0.49 | 68.027 | 2.962 | 0.194 | 0.021 |
| LC-01-08 | 10.6 | 0.46 | 34.602 | 1.437 | 0.523 | 0.021 |
| LC-01-09 | 6.83 | 0.09 | 71.994 | 2.954 | 0.119 | 0.012 |
| LC-01-10 | 13.8 | 0.39 | 74.963 | 2.248 | 0.119 | 0.010 |
| LC-01-11 | 17.9 | 0.20 | 72.202 | 2.085 | 0.309 | 0.019 |
| LC-01-12 | 2.33 | 0.05 | 82.440 | 5.505 | 0.071 | 0.019 |
| LC-01-13 | 5.35 | 0.15 | 79.177 | 3.824 | 0.059 | 0.011 |
| LC-01-14 | 140 | 3.90 | 83.893 | 1.478 | 0.049 | 0.002 |
| LC-01-15 | 16.4 | 0.28 | 39.526 | 2.031 | 0.454 | 0.020 |
| LC-01-16 | 18.3 | 0.51 | 74.239 | 2.645 | 0.108 | 0.010 |
| LC-01-17 | 12.7 | 0.17 | 83.612 | 2.657 | 0.075 | 0.007 |
| LC-01-18 | 10.9 | 0.17 | 84.746 | 2.873 | 0.058 | 0.007 |
| LC-01-19 | 30.5 | 0.39 | 80.645 | 1.886 | 0.097 | 0.006 |
| LC-01-20 | 18.1 | 0.59 | 83.264 | 2.288 | 0.057 | 0.004 |
| LC-01-21 | 14.2 | 0.18 | 81.633 | 2.399 | 0.065 | 0.006 |
| LC-01-22 | 6.77 | 0.48 | 42.373 | 16.88 | 0.127 | 0.031 |
| LC-01-23 | 10.4 | 0.12 | 77.042 | 2.552 | 0.102 | 0.011 |
| LC-01-24 | 12.2 | 0.54 | 41.152 | 1.863 | 0.433 | 0.022 |
| LC-01-25 | 22.8 | 0.24 | 66.050 | 1.614 | 0.239 | 0.012 |
| LC-01-26 | 14.6 | 0.34 | 68.353 | 2.803 | 0.183 | 0.021 |
| LC-01-27 | 8.72 | 0.58 | 77.700 | 3.381 | 0.090 | 0.011 |
| LC-01-28 | 10.2 | 0.73 | 78.555 | 2.962 | 0.087 | 0.009 |
| LC-01-29 | 28.6 | 0.43 | 80.386 | 2.068 | 0.064 | 0.006 |
| LC-01-30 | 18.7 | 0.22 | 71.633 | 2.720 | 0.146 | 0.019 |
| LC-01-31 | 6.53 | 0.10 | 79.554 | 3.164 | 0.060 | 0.008 |
| LC-01-32 | 4.78 | 0.06 | 76.453 | 3.682 | 0.141 | 0.017 |
| LC-01-33 | 31.7 | 0.43 | 82.919 | 1.856 | 0.059 | 0.004 |
| LC-01-34 | 8.89 | 0.12 | 83.682 | 3.011 | 0.050 | 0.005 |
| LC-01-35 | 17.7 | 0.37 | 76.453 | 2.163 | 0.158 | 0.011 |
| LC-01-36 | 8.86 | 0.51 | 69.735 | 2.431 | 0.204 | 0.017 |
| LC-01-37 | 17.1 | 0.33 | 79.365 | 2.079 | 0.081 | 0.006 |
| LC-01-38 | 35.0 | 0.60 | 82.508 | 1.838 | 0.057 | 0.003 |
| LC-01-39 | 22.1 | 0.20 | 75.245 | 2.038 | 0.114 | 0.009 |
| LC-01-40 | 6.95 | 0.11 | 62.189 | 2.746 | 0.258 | 0.021 |
| LC-01-41 | 31.8 | 0.34 | 79.681 | 1.841 | 0.084 | 0.005 |
| LC-01-42 | 28.4 | 0.24 | 64.851 | 1.346 | 0.234 | 0.008 |
| LC-01-43 | 22.5 | 0.73 | 61.996 | 2.537 | 0.226 | 0.026 |

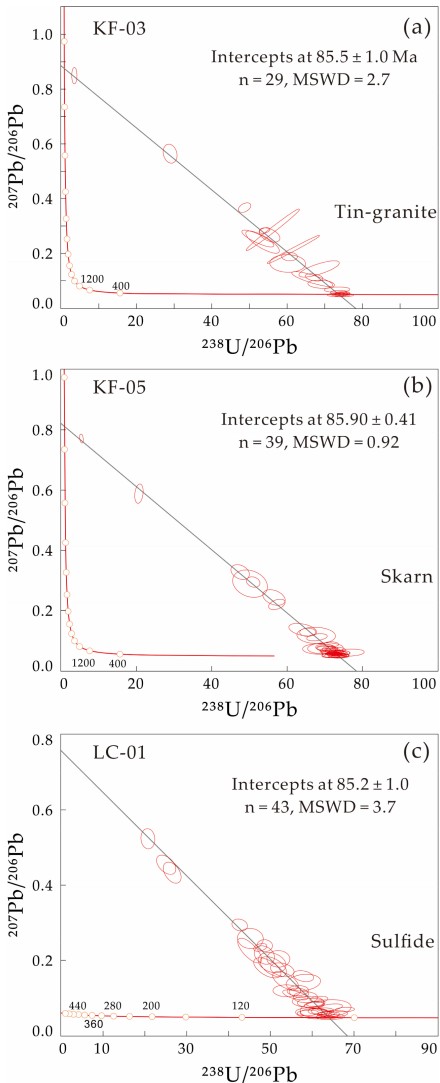

**Figure 5.** Tera–Wasserburg U–Pb plots for cassiterites from the Gejiu tin polymetallic deposit (the ages are calculated at 95% confidence level). (**a**) Plots reveal a mineralization age at $85.5 \pm 1.0$ Ma; (**b**) Plots reveal a mineralization age at $85.90 \pm 0.41$ Ma; (**c**) Plots reveal a mineralization age at $85.2 \pm 1.0$ Ma.

*5.3. Trace Element Composition and Mapping*

Cassiterite grains from different types of cassiterite-bearing ores were analyzed for trace elements, and at least 20 analyses were performed on every cassiterite sample. The analysis results of in situ trace elements for cassiterites are listed in Table 2. The most abundant trace elements in the analyzed cassiterites were Fe (40.73 to >8400 ppm), W (0 to >8500 ppm), Ti (14 to >8400 ppm), and Sc (0–727 ppm). Nb (0–1616 ppm), Ta (0–523 ppm), V (0–822 ppm), Zr (0–483 ppm), Sb (0–229 ppm), and U (0–67.8 ppm) concentrations were also relatively high and vary over several orders of magnitude, even within a single sample of an ore type. Co (8.91–11.96 ppm), Ni (63.1–89.46 ppm) concentrations are relatively constant and Ni/Co ratios (6.59–8.10) vary slightly. Other elements such as Cd, Cu, Zn, Ag, Mo, Rb, Sr, Th, Pb, and REEs were consistently determined to be close to or below detection limits. Hf concentrations were generally below 20 ppm (average value = 4.17 ppm), with U concentrations below 68 ppm.

**Table 2.** Trace element composition of tin-granite, skarn, and massive sulfide types of cassiterite samples analyzed by LA–ICP–MS single spot analysis.

| | SnO$_2$ | FeO$^T$ | TiO$_2$ | SiO$_2$ | Al$_2$O$_3$ | Sc | V | Cr | Co | Ni | Ga | Zr | Nb | Sb | Hf | Ta | W | U |
|---|---|---|---|---|---|---|---|---|---|---|---|---|---|---|---|---|---|---|
| Spot No. | | | wt% | | | | | | | | ppm | | | | | | | |
| **Tin-granite type** | | | | | | | | | | | | | | | | | | |
| KF-03-01 | 99.1 | 0.120 | 0.460 | 0.150 | 0.0019 | 423 | 105 | 3.10 | 10.4 | 75.9 | 2.71 | 133 | 212 | 2.27 | 8.35 | 33.9 | 4.62 | 1.73 |
| KF-03-02 | 99.0 | 0.210 | 0.440 | 0.120 | 0.0021 | 417 | 95.7 | 3.00 | 10.8 | 78.1 | 2.95 | 144 | 944 | 2.87 | 9.90 | 45.1 | 3.57 | 2.76 |
| KF-03-03 | 98.2 | 0.038 | 1.36 | 0.130 | 0.0005 | 63.9 | 2.60 | 2.60 | 10.2 | 81.2 | 0.380 | 72.2 | 349 | 30.0 | 4.77 | 40.8 | 1294 | 10.0 |
| KF-03-04 | 98.7 | 0.045 | 0.680 | 0.130 | 0.0007 | 92.4 | 14.2 | 3.09 | 10.1 | 77.3 | 0.700 | 176 | 787 | 13.1 | 9.50 | 101 | 1919 | 41.5 |
| KF-03-05 | 98.5 | 0.150 | 1.04 | 0.150 | 0.0029 | 313 | 59.9 | 12.20 | 9.58 | 75.5 | 4.08 | 73.6 | 446 | 5.02 | 4.19 | 36.3 | 3.37 | 3.62 |
| KF-03-06 | 98.8 | 0.120 | 0.770 | 0.150 | 0.0051 | 163 | 34.1 | 2.07 | 10.3 | 79.9 | 3.84 | 159 | 577 | 2.39 | 9.51 | 67.6 | 28.1 | 4.32 |
| KF-03-07 | 98.6 | 0.042 | 0.830 | 0.150 | 0.0009 | 137 | 16.5 | 2.50 | 9.82 | 79.6 | 0.72 | 163 | 719 | 13.6 | 8.27 | 105 | 1695 | 35.4 |
| KF-03-08 | 99.1 | 0.081 | 0.490 | 0.160 | 0.0034 | 254 | 39.7 | 4.97 | 10.1 | 77.5 | 3.20 | 138 | 548 | 1.31 | 8.91 | 64.5 | 13.9 | 2.66 |
| KF-03-09 | 98.9 | 0.036 | 0.640 | 0.150 | 0.0006 | 48.5 | 7.36 | 0.06 | 10.3 | 71.8 | 0.750 | 185 | 299 | 37.9 | 10.5 | 32.4 | 1208 | 49.9 |
| KF-03-10 | 99.2 | 0.091 | 0.350 | 0.140 | 0.0009 | 276 | 64.5 | 6.96 | 9.74 | 78.7 | 1.49 | 108 | 624 | 1.49 | 8.94 | 80.9 | 2.61 | 0.880 |
| KF-03-11 | 98.7 | 0.044 | 0.750 | 0.130 | 0.0007 | 98.4 | 14.6 | 3.13 | 10.1 | 74.3 | 0.750 | 174 | 829 | 14.7 | 9.52 | 130 | 1599 | 37.3 |
| KF-03-12 | 98.6 | 0.042 | 0.830 | 0.140 | 0.0007 | 155 | 18.1 | 1.43 | 9.72 | 73.8 | 0.860 | 156 | 731 | 13.8 | 7.32 | 75.1 | 1908 | 34.2 |
| KF-03-13 | 98.0 | 0.052 | 1.51 | 0.130 | 0.0006 | 46.4 | 4.53 | 1.31 | 10.1 | 71.4 | 0.610 | 68.1 | 996 | 15.3 | 4.16 | 154 | 805 | 6.58 |
| KF-03-14 | 99.1 | 0.130 | 0.470 | 0.140 | 0.0014 | 257 | 71.7 | 5.60 | 9.95 | 72.9 | 2.28 | 118 | 366 | 2.21 | 8.51 | 50.8 | 1.39 | 1.44 |
| KF-03-15 | 98.0 | 0.093 | 1.23 | 0.150 | 0.0008 | 65.1 | 15.6 | 4.73 | 9.83 | 70.6 | 1.25 | 138 | 1112 | 30.1 | 7.25 | 119 | 2378 | 38.3 |
| KF-03-16 | 98.2 | 0.074 | 1.23 | 0.150 | 0.0007 | 47.7 | 8.70 | 1.47 | 10.1 | 74.7 | 1.45 | 148 | 889 | 40.6 | 7.55 | 62.2 | 1366 | 38.8 |
| KF-03-17 | 99.3 | 0.078 | 0.320 | 0.130 | 0.0009 | 186 | 49.2 | 6.25 | 10.0 | 71.5 | 1.42 | 101 | 431 | 1.47 | 8.46 | 71.7 | 1.33 | 0.700 |
| KF-03-18 | 98.7 | 0.048 | 0.650 | 0.110 | 0.0007 | 67.1 | 10.3 | — | 9.93 | 78.8 | 0.530 | 171 | 314 | 15.0 | 9.40 | 78.0 | 2563 | 46.3 |
| KF-03-19 | 98.8 | 0.130 | 0.720 | 0.120 | 0.0013 | 121 | 164 | 12.70 | 9.91 | 73.2 | 2.85 | 104 | 913 | 3.23 | 8.40 | 55.7 | 5.57 | 1.23 |
| KF-03-20 | 99.1 | 0.098 | 0.520 | 0.130 | 0.0028 | 314 | 93.0 | 7.58 | 10.1 | 75.7 | 3.29 | 142 | 169 | 1.89 | 8.74 | 21.3 | 2.00 | 1.58 |
| KF-03-21 | 99.2 | 0.100 | 0.460 | 0.130 | 0.0051 | 208 | 40.3 | 5.14 | 10.2 | 75.4 | 5.07 | 97.5 | 257 | 1.96 | 6.34 | 52.6 | 19.0 | 2.70 |
| KF-03-22 | 97.4 | 0.062 | 1.98 | 0.140 | 0.0009 | 26.7 | 4.51 | 4.14 | 9.74 | 73.5 | 0.89 | 73.5 | 538 | 12.7 | 3.97 | 88.7 | 2503 | 12.1 |
| KF-03-23 | 99.0 | 0.059 | 0.690 | 0.120 | 0.0009 | 119 | 37.9 | 14.0 | 9.97 | 72.7 | 1.31 | 68.5 | 332 | 2.13 | 6.13 | 35.6 | 0.87 | 0.290 |
| KF-03-24 | 98.1 | 0.082 | 1.23 | 0.130 | 0.0006 | 60.8 | 13.2 | 5.97 | 9.63 | 74.0 | 1.10 | 153 | 1192 | 41.0 | 8.22 | 110 | 1579 | 42.6 |
| KF-03-25 | 99.2 | 0.078 | 0.320 | 0.130 | 0.0006 | 158 | 95.6 | 10.30 | 10.1 | 75.6 | 0.910 | 84.5 | 832 | 1.80 | 11.7 | 236 | 2.83 | 0.530 |
| **Skarn type** | | | | | | | | | | | | | | | | | | |
| KF-05-01 | 98.8 | 0.830 | 0.150 | 0.114 | 0.019 | 44.0 | 6.88 | — | 10.3 | 78.5 | 26.4 | 36.0 | 206 | 43 | 3.76 | 15.1 | 119 | 18.7 |
| KF-05-02 | 99.0 | 0.550 | 0.160 | 0.113 | 0.0093 | 239 | 21.6 | 1.74 | 10.0 | 76.4 | 17.6 | 18.5 | 340 | 46 | 2.01 | 21.7 | 84.9 | 6.08 |
| KF-05-03 | 99.5 | 0.130 | 0.110 | 0.105 | 0.0041 | 50.9 | 10.04 | 0.34 | 10.3 | 80.7 | 5.29 | 37.9 | 300 | 4.7 | 4.97 | 133 | 97.0 | 2.06 |
| KF-05-04 | 99.4 | 0.035 | 0.026 | 0.122 | 0.0018 | 0.230 | 0.024 | — | 11.3 | 83.8 | 2.17 | 23.2 | 89.1 | 22 | 1.84 | 6.75 | 2823 | 38.0 |
| KF-05-05 | 98.8 | 0.820 | 0.093 | 0.16 | 0.0200 | 37.0 | 7.15 | 0.01 | 11.0 | 89.2 | 26.4 | 48.4 | 31.6 | 18.5 | 5.06 | 3.66 | 29.1 | 8.37 |
| KF-05-06 | 98.9 | 0.790 | 0.092 | 0.140 | 0.0200 | 37.5 | 7.39 | — | 12.0 | 89.5 | 25.7 | 44.9 | 27.3 | 17.2 | 5.73 | 1.68 | 44.7 | 8.27 |
| KF-05-07 | 99.1 | 0.170 | 0.036 | 0.123 | 0.0130 | 10.4 | 1.39 | 0.05 | 11.7 | 82.5 | 6.58 | 25.6 | 547 | 11.8 | 2.93 | 78.1 | 3873 | 15.7 |
| KF-05-08 | 99.0 | 0.770 | 0.030 | 0.119 | 0.0190 | 21.8 | 4.13 | — | 11.1 | 81.6 | 25.0 | 1.76 | 12.3 | 17.5 | 0.35 | 2.97 | 19.0 | 7.93 |
| KF-05-09 | 99.2 | 0.160 | 0.330 | 0.125 | 0.0029 | 120 | 14.2 | 1.76 | 11.0 | 81.4 | 4.60 | 96.0 | 378 | 3.3 | 12.6 | 148 | 4.36 | 0.960 |
| KF-05-10 | 98.2 | 0.580 | 0.710 | 0.140 | 0.0126 | 281 | 122 | 6.15 | 10.6 | 81.8 | 15.5 | 116 | 1175 | 13.3 | 11.4 | 83.1 | 48.2 | 10.6 |
| KF-05-11 | 98.9 | 0.690 | 0.130 | 0.132 | 0.0140 | 175 | 28.9 | 1.10 | 10.8 | 78.8 | 25.5 | 12.3 | 86.5 | 31 | 1.42 | 50.8 | 22.6 | 5.92 |
| KF-05-12 | 99.1 | 0.170 | 0.033 | 0.118 | 0.0124 | 3.14 | 1.25 | 0.360 | 11.0 | 77.9 | 8.73 | 14.4 | 180 | 11.8 | 1.70 | 53.5 | 3958 | 19.2 |
| KF-05-13 | 98.9 | 0.800 | 0.049 | 0.107 | 0.0180 | 15.2 | 2.20 | 0.360 | 10.8 | 79.7 | 23.9 | 10.6 | 53.8 | 51 | 0.76 | 0.440 | 122 | 25.3 |
| KF-05-14 | 98.7 | 0.630 | 0.300 | 0.140 | 0.0130 | 96.4 | 25.0 | 1.17 | 10.4 | 78.8 | 18.0 | 66.9 | 715 | 24 | 7.15 | 114 | 87.4 | 14.5 |
| KF-05-15 | 98.4 | 0.320 | 0.450 | 0.130 | 0.0076 | 727 | 371 | 6.82 | 11.0 | 83.2 | 7.38 | 171 | 1616 | 10.7 | 19.3 | 55.7 | 1592 | 14.5 |
| KF-05-16 | 99.3 | 0.360 | 0.103 | 0.119 | 0.0096 | 155 | 34.5 | 1.10 | 10.7 | 75.5 | 10.9 | 103 | 83.9 | 13.5 | 13.3 | 2.30 | 13.6 | 5.95 |
| KF-05-17 | 99.0 | 0.640 | 0.150 | 0.122 | 0.0140 | 44.3 | 9.78 | — | 11.6 | 80.4 | 20.2 | 41.9 | 149 | 20 | 5.32 | 4.42 | 62.9 | 8.85 |
| KF-05-18 | 99.0 | 0.830 | 0.014 | 0.102 | 0.0190 | 10.1 | 0.72 | 0.920 | 10.5 | 81.7 | 32.0 | 1.47 | 18.0 | 38 | 0.14 | 2.74 | 61.0 | 13.9 |
| KF-05-19 | 99.4 | 0.160 | 0.170 | 0.116 | 0.0046 | 28.1 | 6.66 | 1.98 | 10.5 | 83.6 | 6.19 | 25.0 | 266 | 2.7 | 3.76 | 252 | 6.85 | 1.57 |
| KF-05-20 | 98.0 | 0.490 | 0.960 | 0.132 | 0.0115 | 370 | 132 | 21.2 | 11.0 | 78.4 | 13.0 | 119 | 1474 | 12.5 | 12.6 | 327 | 39.3 | 7.32 |
| KF-05-21 | 99.6 | 0.160 | 0.027 | 0.140 | 0.0048 | 3.12 | 0.840 | 0.420 | 11.1 | 82.8 | 6.38 | 37.3 | 44.2 | 3.0 | 4.64 | 12.7 | 8.88 | 1.76 |
| KF-05-22 | 98.9 | 0.520 | 0.290 | 0.117 | 0.0109 | 167 | 16.3 | 1.36 | 11.1 | 81.0 | 14.8 | 32.5 | 385 | 12.8 | 3.39 | 20.5 | 10.3 | 5.50 |
| KF-05-23 | 99.5 | 0.170 | 0.062 | 0.120 | 0.0040 | 682 | 51.5 | — | 11.5 | 78.4 | 4.75 | 53.2 | 115 | 11.6 | 5.89 | 19.8 | 6.58 | 1.15 |
| KF-05-24 | 99.0 | 0.390 | 0.260 | 0.120 | 0.0050 | 191 | 33.1 | 1.71 | 10.8 | 77.7 | 11.1 | 28.0 | 626 | 36 | 4.34 | 155 | 88.7 | 5.47 |
| KF-05-25 | 99.0 | 0.500 | 0.140 | 0.118 | 0.0131 | 602 | 27.8 | 0.290 | 11.0 | 76.4 | 16.8 | 25.1 | 236 | 23 | 2.95 | 77.0 | 75.7 | 3.15 |
| KF-05-26 | 99.3 | 0.330 | 0.160 | 0.099 | 0.0060 | 178 | 21.4 | — | 10.7 | 79.9 | 10.9 | 26.4 | 225 | 21 | 3.12 | 92.7 | 39.4 | 3.08 |
| KF-05-27 | 99.5 | 0.300 | 0.021 | 0.120 | 0.0126 | 4.95 | 2.47 | 0.270 | 10.9 | 77.6 | 15.8 | 10.2 | 43.7 | 7.4 | 1.03 | 10.5 | 28.7 | 3.34 |
| KF-05-28 | 99.0 | 0.770 | 0.042 | 0.125 | 0.0190 | 42.4 | 3.12 | 0.160 | 11.1 | 78.5 | 30.6 | 13.5 | 40.0 | 34 | 1.61 | 15.0 | 42.9 | 5.95 |
| KF-05-29 | 98.9 | 0.690 | 0.170 | 0.119 | 0.0120 | 447 | 26.8 | 0.700 | 10.5 | 82.2 | 22.2 | 14.5 | 204 | 33 | 1.99 | 3.73 | 24.5 | 5.35 |
| KF-05-30 | 98.7 | 0.750 | 0.340 | 0.115 | 0.0170 | 63.7 | 17.9 | 2.52 | 10.3 | 80.7 | 23.0 | 55.3 | 212 | 16.6 | 5.73 | 62.9 | 27.8 | 9.70 |
| KF-05-31 | 98.6 | 0.700 | 0.430 | 0.129 | 0.0140 | 31.4 | 10.97 | 1.53 | 10.7 | 80.7 | 20.8 | 62.3 | 232 | 14.4 | 6.95 | 87.3 | 25.9 | 10.7 |
| KF-05-32 | 99.0 | 0.560 | 0.140 | 0.126 | 0.0084 | 233 | 16.7 | 0.390 | 11.1 | 79.4 | 16.3 | 20.2 | 413 | 54 | 2.77 | 32.8 | 82.3 | 7.22 |
| KF-05-33 | 99.6 | 0.036 | 0.170 | 0.119 | 0.0009 | 72.2 | 9.79 | 0.820 | 10.7 | 77.5 | 1.12 | 22.9 | 135 | 2.1 | 4.18 | 197 | 12.0 | 0.13 |
| KF-05-34 | 98.8 | 0.640 | 0.340 | 0.115 | 0.0137 | 133 | 43.7 | 1.14 | 11.2 | 77.5 | 17.8 | 18.6 | 295 | 12.8 | 1.87 | 19.6 | 15.5 | 8.58 |
| KF-05-35 | 98.9 | 0.830 | 0.095 | 0.118 | 0.0210 | 58.4 | 37.3 | 0.930 | 11.1 | 78.8 | 25.0 | 7.64 | 51.1 | 21 | 0.89 | 8.77 | 26.7 | 10.1 |
| KF-05-36 | 99.4 | 0.200 | 0.160 | 0.132 | 0.0046 | 158 | 13.8 | 2.07 | 11.1 | 76.9 | 6.36 | 61.9 | 352 | 6.7 | 7.35 | 174 | 10.2 | 2.59 |
| KF-05-37 | 99.3 | 0.340 | 0.105 | 0.120 | 0.0080 | 114 | 8.53 | 1.26 | 10.7 | 79.9 | 13.6 | 28.6 | 443 | 21 | 3.66 | 81.7 | 19.1 | 5.16 |
| KF-05-38 | 98.8 | 0.840 | 0.096 | 0.118 | 0.0200 | 61.7 | 12.3 | — | 10.7 | 77.3 | 26.2 | 24.9 | 46.0 | 23 | 2.75 | 1.65 | 33.2 | 11.6 |
| KF-05-39 | 98.9 | 0.870 | 0.056 | 0.110 | 0.0220 | 37.0 | 12.6 | 1.86 | 10.9 | 79.9 | 28.2 | 30.4 | 17.3 | 20 | 3.69 | 1.20 | 25.8 | 9.06 |
| **Massive sulfide type** | | | | | | | | | | | | | | | | | | |
| LC-01-01 | 98.9 | 0.160 | 0.011 | 0.820 | 0.0044 | 22.0 | 42.3 | — | 10.5 | 75.6 | 1.61 | 60.4 | 12.0 | 9.10 | 3.01 | 3.47 | 2.81 | 0.940 |
| LC-01-02 | 93.0 | 0.730 | 0.020 | 4.11 | 1.3200 | 56.3 | 58.6 | 0.970 | 9.71 | 71.6 | 8.94 | 50.4 | 461 | 90.9 | 2.34 | 57.6 | 2345 | 17.9 |
| LC-01-03 | 98.8 | 0.460 | 0.021 | 0.810 | 0.0190 | 72.4 | 79.8 | 0.530 | 10.4 | 75.5 | 5.42 | 35.7 | 36.4 | 36.4 | 1.59 | 54.4 | 14.2 | 5.35 |
| LC-01-04 | 99.0 | 0.140 | 0.012 | 0.710 | 0.0081 | 19.3 | 22.4 | 0.480 | 10.3 | 73.6 | 1.12 | 68.0 | 427 | 18.3 | 3.80 | 62.9 | 92.2 | 3.29 |
| LC-01-05 | 86.1 | 1.210 | 0.026 | 8.57 | 2.9000 | 79.5 | 109 | 1.50 | 8.98 | 68.8 | 12.7 | 80.1 | 914 | 55.6 | 4.74 | 146 | 900 | 11.9 |
| LC-01-06 | 98.9 | 0.150 | 0.009 | 0.79 | 0.0047 | 28.4 | 77.9 | 0.600 | 10.6 | 79.7 | 1.58 | 69.1 | 231 | 17.4 | 3.28 | 28.4 | 524 | 5.00 |
| LC-01-07 | 98.1 | 0.660 | 0.039 | 0.760 | 0.0210 | 131 | 69.9 | 2.27 | 10.2 | 77.8 | 5.97 | 141 | 1400 | 79.0 | 9.29 | 523 | 106 | 14.6 |
| LC-01-08 | 98.6 | 0.450 | 0.017 | 0.780 | 0.0160 | 132 | 222 | 1.17 | 10.5 | 76.7 | 5.44 | 105 | 113 | 39.2 | 6.30 | 44.0 | 15.2 | 6.36 |
| LC-01-09 | 98.7 | 0.170 | 0.016 | 0.730 | 0.0120 | 17.9 | 18.6 | 0.390 | 10.3 | 74.6 | 1.63 | 132 | 132 | 24.8 | 0.99 | 10.9 | 2514 | 11.0 |
| LC-01-10 | 98.7 | 0.150 | 0.008 | 0.860 | 0.0070 | 16.8 | 17.2 | 0.800 | 11.2 | 79.8 | 1.61 | 37.4 | 183 | 51.3 | 1.68 | 38.1 | 1647 | 10.2 |
| LC-01-11 | 98.5 | 0.260 | 0.025 | 0.840 | 0.0100 | 67.8 | 121 | 2.25 | 10.1 | 76.0 | 2.97 | 77.0 | 1379 | 23.3 | 5.50 | 224 | 555 | 8.86 |
| LC-01-12 | 98.3 | 0.710 | 0.024 | 0.740 | 0.0220 | 86.9 | 72.3 | 1.64 | 10.4 | 75.6 | 8.73 | 61.4 | 395 | 121 | 3.21 | 26.9 | 72.9 | 14.4 |
| LC-01-13 | 98.8 | 0.140 | 0.021 | 0.750 | 0.0069 | 14.8 | 16.1 | — | 10.7 | 76.8 | 1.41 | 37.4 | 135 | 48.3 | 1.85 | 32.7 | 1842 | 9.26 |
| LC-01-14 | 98.3 | 0.480 | 0.021 | 0.840 | 0.0380 | 63.1 | 54.2 | — | 10.5 | 76.5 | 6.59 | 50.8 | 258 | 90.5 | 2.27 | 12.6 | 1500 | 16.8 |
| LC-01-15 | 98.5 | 0.160 | 0.016 | 0.940 | 0.0064 | 28.9 | 52.4 | — | 10.5 | 76.0 | 1.84 | 55.1 | 206 | 57.9 | 2.28 | 11.2 | 2359 | 11.6 |
| LC-01-16 | 99.3 | 0.170 | 0.050 | 0.092 | 0.0072 | 31.5 | 62.8 | — | 10.6 | 76.6 | 1.44 | 58.7 | 593 | 72.7 | 2.93 | 62.8 | 2143 | 13.3 |
| LC-01-17 | 98.5 | 0.500 | 0.023 | 0.740 | 0.0210 | 161 | 231 | — | 9.97 | 76.7 | 6.99 | 96.1 | 483 | 73.9 | 5.21 | 93.1 | 310 | 9.83 |

**Table 2.** *Cont.*

| Spot No. | SnO$_2$ | FeO$^T$ | TiO$_2$ | SiO$_2$ | Al$_2$O$_3$ | Sc | V | Cr | Co | Ni | Ga | Zr | Nb | Sb | Hf | Ta | W | U |
|---|---|---|---|---|---|---|---|---|---|---|---|---|---|---|---|---|---|---|
| | | | wt% | | | | | | | | | ppm | | | | | | |
| **Massive sulfide type** | | | | | | | | | | | | | | | | | | |
| LC-01-18 | 98.7 | 0.330 | 0.020 | 0.750 | 0.0120 | 154 | 337 | 3.86 | 10.5 | 77.0 | 4.37 | 130 | 339 | 39.0 | 6.82 | 61.3 | 260 | 7.29 |
| LC-01-19 | 98.5 | 0.220 | 0.021 | 0.760 | 0.0130 | 24.5 | 22.7 | 1.15 | 10.2 | 75.1 | 2.30 | 40.2 | 412 | 76.9 | 2.02 | 114 | 2649 | 17.0 |
| LC-01-20 | 98.7 | 0.400 | 0.031 | 0.770 | 0.0110 | 62.4 | 64.8 | 0.930 | 10.1 | 77.2 | 3.61 | 46.6 | 75.3 | 60.2 | 2.08 | 9.70 | 18.1 | 5.52 |
| LC-01-21 | 98.3 | 0.490 | 0.019 | 0.970 | 0.0190 | 180 | 309 | 3.31 | 9.88 | 74.5 | 7.47 | 96.8 | 113 | 78.7 | 4.88 | 11.9 | 58.7 | 8.28 |
| LC-01-22 | 98.2 | 0.510 | 0.022 | 1.03 | 0.0670 | 78.7 | 67.9 | 0.670 | 10.0 | 75.1 | 5.86 | 85.7 | 424 | 54.7 | 4.73 | 128 | 251 | 10.8 |
| LC-01-23 | 81.8 | 1.340 | 0.042 | 11.7 | 3.7600 | 52.0 | 69.8 | 9.62 | 8.91 | 63.1 | 12.0 | 55.5 | 239 | 50.8 | 2.82 | 39.3 | 316 | 6.05 |
| LC-01-24 | 98.4 | 0.400 | 0.050 | 0.940 | 0.0740 | 65.7 | 69.2 | 0.490 | 10.7 | 76.4 | 4.25 | 55.5 | 114 | 57.3 | 2.62 | 11.9 | 35.0 | 5.62 |
| LC-01-25 | 98.9 | 0.300 | 0.015 | 0.630 | 0.0100 | 177 | 385 | 1.75 | 10.6 | 77.7 | 3.83 | 134 | 67.2 | 29.0 | 7.31 | 24.1 | 25.7 | 4.10 |
| LC-01-26 | 98.3 | 0.180 | 0.023 | 0.980 | 0.0810 | 16.8 | 19.0 | — | 9.85 | 75.7 | 1.41 | 37.5 | 158 | 63.7 | 1.78 | 15.6 | 2357 | 11.8 |
| LC-01-27 | 98.7 | 0.400 | 0.023 | 0.650 | 0.0170 | 90.4 | 141 | 4.17 | 10.3 | 77.9 | 4.66 | 85.7 | 442 | 58.9 | 5.08 | 107 | 290 | 8.74 |
| LC-01-28 | 98.3 | 0.270 | 0.022 | 0.740 | 0.0250 | 35.7 | 33.1 | 1.27 | 10.2 | 72.8 | 3.23 | 39.6 | 483 | 128 | 2.10 | 69.1 | 3783 | 17.7 |
| LC-01-29 | 98.8 | 0.320 | 0.014 | 0.810 | 0.0078 | 41.7 | 43.3 | 2.34 | 10.0 | 79.0 | 3.02 | 50.2 | 32.8 | 33.6 | 1.87 | 1.45 | 12.6 | 3.65 |
| LC-01-30 | 98.9 | 0.290 | 0.023 | 0.750 | 0.0074 | 41.7 | 41.6 | — | 10.7 | 76.1 | 2.63 | 59.9 | 35.6 | 29.1 | 2.17 | 2.84 | 16.9 | 3.37 |
| LC-01-31 | 98.9 | 0.098 | 0.010 | 0.780 | 0.0046 | 9.43 | 12.5 | 1.75 | 10.3 | 77.6 | 0.770 | 40.7 | 265 | 27.6 | 2.28 | 72.0 | 912 | 5.40 |
| LC-01-32 | 98.2 | 0.270 | 0.024 | 0.850 | 0.0270 | 33.3 | 36.4 | 0.440 | 10.3 | 78.4 | 3.78 | 47.3 | 661 | 80.3 | 2.00 | 55.4 | 3180 | 35.7 |
| LC-01-33 | 98.9 | 0.140 | 0.007 | 0.720 | 0.0081 | 13.6 | 15.4 | — | 10.7 | 79.7 | 1.52 | 36.6 | 150 | 43.7 | 1.46 | 40.8 | 1295 | 8.10 |
| LC-01-34 | 98.9 | 0.110 | 0.009 | 0.730 | 0.0055 | 11.2 | 15.2 | 1.70 | 10.7 | 76.9 | 1.07 | 38.7 | 207 | 29.0 | 2.17 | 82.8 | 991 | 5.45 |
| LC-02-01 | 97.0 | 0.730 | 0.340 | 0.780 | 0.0240 | 144 | 440 | 6.74 | 10.5 | 73.9 | 6.38 | 101 | 170 | 229 | 3.73 | 2.73 | 7366 | 58.2 |
| LC-02-02 | 98.3 | 0.091 | 0.430 | 0.690 | 0.0084 | 13.2 | 30.5 | 5.72 | 10.2 | 76.9 | 1.76 | 54.6 | 78.0 | 27.3 | 1.98 | 6.56 | 3209 | 15.7 |
| LC-02-03 | 98.1 | 0.100 | 0.440 | 0.790 | 0.0057 | 13.6 | 30.1 | 0.04 | 10.3 | 74.2 | 2.25 | 59.7 | 56.8 | 22.0 | 1.71 | 2.59 | 4147 | 19.1 |
| LC-02-04 | 97.5 | 0.230 | 0.220 | 1.14 | 0.0700 | 45.4 | 85.3 | 0.440 | 9.94 | 76.8 | 6.09 | 30.8 | 138 | 43.5 | 1.00 | 3.29 | 5796 | 15.7 |
| LC-02-05 | 97.5 | 0.400 | 0.100 | 0.860 | 0.0340 | 27.7 | 148 | 1.98 | 10.4 | 75.2 | 6.74 | 15.4 | 8.27 | 84.7 | 0.27 | 0.017 | 7676 | 26.9 |
| LC-02-06 | 98.3 | 0.670 | 0.160 | 0.580 | 0.0320 | 111 | 723 | 2.87 | 10.2 | 74.4 | 12.0 | 56.6 | 4.73 | 71.6 | 1.68 | 0.14 | 337 | 13.1 |
| LC-02-07 | 98.4 | 0.130 | 0.290 | 0.700 | 0.0130 | 20.3 | 38.0 | 0.950 | 10.3 | 74.8 | 1.51 | 30.8 | 68.8 | 26.1 | 0.82 | 1.58 | 3325 | 12.1 |
| LC-02-08 | 98.1 | 0.093 | 0.400 | 0.830 | 0.0080 | 21.9 | 32.9 | 0.10 | 10.6 | 72.5 | 2.10 | 34.9 | 71.7 | 22.5 | 1.03 | 2.47 | 3800 | 19.3 |
| LC-02-09 | 96.0 | 1.250 | 0.910 | 1.15 | 0.2000 | 380 | 530 | 4.65 | 10.3 | 74.1 | 13.3 | 483 | 268 | 94.6 | 24.4 | 28.1 | 194 | 40.2 |
| LC-02-10 | 98.8 | 0.038 | 0.280 | 0.750 | 0.0004 | 34.2 | 45.2 | — | 10.9 | 76.5 | 0.420 | 78.1 | 433 | 4.31 | 4.82 | 26.3 | 54.3 | 3.17 |
| LC-02-11 | 98.2 | 0.160 | 0.250 | 0.850 | 0.0230 | 28.2 | 77.4 | 0.740 | 11.0 | 72.4 | 2.45 | 51.9 | 98.7 | 32.1 | 2.14 | 4.11 | 3371 | 11.8 |
| LC-02-12 | 96.3 | 0.700 | 1.53 | 0.790 | 0.0140 | 121 | 510 | 25.70 | 9.91 | 73.7 | 6.59 | 207 | 235 | 118 | 12.3 | 18.6 | 3029 | 67.8 |
| LC-02-13 | 97.3 | 0.280 | 0.880 | 0.690 | 0.0230 | 32.2 | 108 | 4.73 | 10.7 | 75.2 | 3.42 | 81.1 | 162 | 34.9 | 4.12 | 30.1 | 6071 | 25.4 |
| LC-02-14 | 98.5 | 0.520 | 0.094 | 0.740 | 0.0074 | 19.3 | 35.2 | — | 10.5 | 74.7 | 1.76 | 100 | 91.9 | 48.5 | 3.63 | 6.54 | 16.4 | 8.82 |
| LC-02-15 | 97.4 | 0.740 | 0.043 | 0.800 | 0.0270 | 22.7 | 65.1 | — | 10.2 | 77.6 | 9.08 | 49.9 | 2.12 | 127 | 1.01 | 0.0081 | 6964 | 39.8 |
| LC-02-16 | 98.4 | 0.170 | 0.480 | 0.770 | 0.0120 | 39.6 | 121 | 1.73 | 10.4 | 77.7 | 3.29 | 55.9 | 32.0 | 22.6 | 1.83 | 1.03 | 808 | 8.27 |
| LC-02-17 | 98.6 | 0.300 | 0.160 | 0.730 | 0.0190 | 118 | 509 | 4.24 | 10.2 | 75.6 | 6.88 | 82.1 | 2.24 | 25.8 | 2.61 | 0.076 | 26.1 | 5.34 |
| LC-02-18 | 98.3 | 0.840 | 0.024 | 0.670 | 0.0200 | 63.1 | 111 | 1.89 | 10.6 | 73.8 | 9.53 | 17.3 | 30.9 | 150 | 0.22 | 0.030 | 111 | 25.9 |
| LC-02-19 | 91.9 | 1.220 | 0.540 | 5.88 | 0.1700 | 141 | 331 | — | 9.79 | 69.5 | 23.5 | 56.6 | 14.9 | 145 | 1.42 | 0.500 | 660 | 30.0 |
| LC-02-20 | 98.1 | 0.770 | 0.180 | 0.770 | 0.0690 | 53.7 | 215 | — | 10.1 | 74.8 | 23.7 | 47.8 | 3.29 | 72.9 | 1.30 | 0.047 | 362 | 11.7 |
| LC-02-21 | 97.3 | 1.170 | 0.380 | 0.860 | 0.0190 | 143 | 648 | 3.03 | 10.1 | 73.8 | 8.97 | 69.3 | 27.8 | 188 | 2.62 | 1.19 | 158 | 33.8 |
| LC-02-22 | 98.2 | 0.780 | 0.022 | 0.870 | 0.0180 | 42.9 | 87.0 | 0.790 | 10.0 | 73.5 | 7.64 | 28.0 | 15.1 | 117 | 0.61 | 0.034 | 70.7 | 18.2 |
| LC-02-24 | 96.8 | 0.960 | 1.22 | 0.780 | 0.0170 | 131 | 484 | 6.37 | 10.5 | 71.3 | 9.31 | 187 | 129 | 68.3 | 8.54 | 8.40 | 183 | 31.6 |
| LC-02-25 | 98.3 | 0.100 | 0.300 | 0.850 | 0.0098 | 15.7 | 52.1 | — | 10.6 | 75.9 | 1.13 | 46.1 | 62.2 | 10.5 | 1.28 | 0.730 | 3127 | 8.54 |
| LC-02-26 | 98.5 | 0.330 | 0.420 | 0.640 | 0.0160 | 112 | 216 | 3.34 | 10.3 | 71.4 | 4.37 | 38.1 | 18.1 | 21.2 | 0.97 | 0.420 | 133 | 7.24 |
| LC-02-27 | 97.0 | 1.110 | 0.860 | 0.750 | 0.0320 | 242 | 725 | 25.30 | 9.86 | 72.2 | 11.1 | 159 | 42.1 | 62.8 | 9.85 | 5.32 | 37.0 | 27.7 |
| **Oxidized type** | | | | | | | | | | | | | | | | | | |
| SK-04-01 | 99.1 | 0.250 | 0.500 | 0.130 | 0.0230 | 1.35 | 87.0 | 4.36 | 10.8 | 78.9 | 23.8 | 17.7 | 0.790 | 3.24 | 0.36 | 0.029 | 12.4 | 3.26 |
| SK-04-03 | 99.7 | 0.049 | 0.120 | 0.130 | 0.0014 | 34.0 | 37.4 | — | 10.9 | 79.9 | 2.77 | 50.6 | 25.2 | 0.33 | 1.79 | 3.19 | 9.7 | 0.430 |
| SK-04-04 | 99.1 | 0.033 | 0.045 | 0.140 | 0.0012 | 0.380 | 1.12 | 0.320 | 10.8 | 76.8 | 2.86 | 13.9 | 1.93 | 4.11 | 0.42 | 0.0054 | 5122 | 7.05 |
| SK-04-05 | 99.4 | 0.150 | 0.150 | 0.140 | 0.0052 | 11.0 | 20.1 | 4.13 | 10.3 | 79.5 | 12.2 | 30.4 | 0.780 | 3.69 | 0.88 | 0.011 | 796 | 5.17 |
| SK-04-06 | 98.5 | 0.068 | 0.230 | 0.081 | 0.0044 | 1.90 | 4.07 | 0.350 | 10.1 | 76.9 | 10.9 | 6.34 | 5.25 | 9.8 | 0.12 | 0.011 | 8510 | 20.8 |
| SK-04-07 | 99.7 | 0.094 | 0.070 | 0.130 | 0.0106 | 0.50 | 21.8 | 7.71 | 10.8 | 79.7 | 8.53 | 14.9 | 0.130 | 0.78 | 0.58 | 0.018 | 10.6 | 0.310 |
| SK-04-08 | 99.6 | 0.150 | 0.033 | 0.140 | 0.0049 | 2.11 | 7.59 | 1.98 | 10.9 | 79.5 | 16.6 | 4.37 | 0.600 | 5.33 | 0.18 | 0.011 | 473 | 2.50 |
| SK-04-09 | 98.1 | 0.069 | 0.780 | 0.130 | 0.0015 | 2.49 | 26.2 | 15.5 | 10.9 | 75.9 | 11.1 | 24.3 | 39.5 | 10.1 | 0.39 | 0.100 | 7383 | 30.7 |
| SK-04-10 | 99.0 | 0.600 | 0.230 | 0.120 | 0.0150 | 16.2 | 38.8 | 9.27 | 10.7 | 78.8 | 70.5 | 11.4 | 1.04 | 13.1 | 0.24 | 0.010 | 81.2 | 8.37 |
| SK-04-11 | 98.2 | 0.065 | 0.690 | 0.120 | 0.0022 | 3.67 | 25.0 | 8.97 | 10.1 | 75.3 | 27.2 | 22.3 | 36.8 | 13.4 | 0.51 | 0.110 | 6965 | 30.4 |
| SK-04-12 | 99.7 | 0.026 | 0.069 | 0.140 | 0.0004 | 12.0 | 114 | 0.440 | 10.8 | 79.5 | 1.64 | 11.4 | 0.460 | 0.45 | 0.34 | 0.0075 | 4.43 | 0.460 |
| SK-04-13 | 99.2 | 0.100 | 0.009 | 0.120 | 0.0130 | 0.490 | 4.18 | 0.690 | 10.5 | 79.1 | 4.94 | 0.920 | 0.410 | 37.6 | 0.075 | 0.016 | 3841 | 8.14 |
| SK-04-14 | 99.6 | 0.009 | 0.200 | 0.140 | 0.0001 | 1.42 | 97.5 | 83.7 | 10.7 | 77.6 | 0.430 | 84.3 | 7.50 | 0.74 | 4.56 | 2.19 | 205 | 0.470 |
| SK-04-15 | 99.6 | 0.036 | 0.010 | 0.120 | 0.0015 | 0.03 | 0.620 | 0.960 | 10.3 | 78.2 | 26.1 | 0.67 | 0.110 | 25.3 | 0.022 | 0.013 | 1477 | 12.4 |
| SK-05-03 | 98.6 | 0.500 | 0.570 | 0.120 | 0.0120 | 2.55 | 822 | 94.2 | 10.2 | 75.4 | 65.4 | 64.7 | 19.8 | 12.0 | 1.98 | 0.500 | 5.71 | 11.0 |
| SK-05-04 | 99.1 | 0.240 | 0.400 | 0.130 | 0.0084 | 0.680 | 454 | 39.3 | 10.3 | 81.1 | 35.4 | 53.6 | 14.3 | 25.3 | 1.53 | 0.140 | 142 | 26.0 |
| SK-05-06 | 99.2 | 0.580 | 0.034 | 0.140 | 0.0170 | 0.320 | 39.0 | 1.40 | 10.6 | 75.3 | 75.9 | 7.13 | 0.340 | 26.3 | 0.11 | 0.0082 | 35.7 | 8.32 |
| SK-05-07 | 99.3 | 0.400 | 0.038 | 0.130 | 0.0052 | 0.180 | 28.2 | 2.65 | 10.2 | 76.9 | 54.9 | 8.60 | 1.07 | 16.6 | 0.26 | 0.0057 | 501 | 6.68 |
| SK-05-11 | 99.3 | 0.460 | 0.003 | 0.150 | 0.0120 | 0.077 | 1.85 | 3.63 | 10.8 | 78.6 | 87.6 | 9.80 | 0.00470 | 15.3 | 0.27 | 0.0027 | 0.74 | 4.80 |
| SK-05-12 | 99.8 | 0.006 | 0.004 | 0.160 | 0.0003 | 0.004 | 1.53 | 2.13 | 10.4 | 76.1 | 1.16 | 0.750 | 0.0490 | 12.1 | 0.016 | 0.0028 | 178 | 11.6 |
| SK-05-17 | 99.1 | 0.014 | 0.590 | 0.150 | 0.0007 | 0.540 | 65.0 | 61.2 | 10.7 | 78.2 | 2.62 | 156 | 23.5 | 26.5 | 4.25 | 0.230 | 253 | 22.7 |
| SK-05-18 | 99.6 | 0.110 | 0.120 | 0.140 | 0.0042 | 10.7 | 36.9 | 5.35 | 9.89 | 75.7 | 12.2 | 14.5 | 0.490 | 2.47 | 0.35 | 0.009 | 22.0 | 0.690 |
| SK-05-19 | 99.1 | 0.540 | 0.170 | 0.130 | 0.0030 | 40.6 | 115 | — | 10.2 | 75.7 | 72.8 | 16.5 | 2.05 | 10.2 | 0.40 | 0.090 | 121 | 10.2 |

"—" represent below detection limit.

LA–ICP–MS mapping enabled a suite of trace elements to be simultaneously measured across the surface of the cassiterites. It is therefore a powerful tool for understanding the relationship between chemical composition and color variations. Trace element mapping results show a significant relationship between element distribution and CL-defined zoning in all cassiterite grains. The distribution of Nb, Ta, and Ti, revealed by elemental mapping (Figure 6), correlates with the regular oscillatory zoning pattern displayed by CL images. Dark luminescing domains of these cassiterite grains often showed relatively high Sb, Fe, W, Ga, and U, and low Nb and Ta [8,44]. In this study, we discuss skarn (KF–05), and massive sulfide samples (LC–01) in detail as examples. In elemental maps of the cassiterite grain

from skarn ores (see Figure 6a), the dark luminescing domains reveal high Nb, Ta, W, and U contents. The regular oscillatory zoning pattern correlates with the distribution of Nb, Ta, and Ti. However, in elemental maps of the cassiterite grain from massive sulfide ores (see Figure 6b), dark luminescing domains reveal high Sb, W, and U contents. The regular oscillatory zoning pattern correlates with the distribution of Nb, Ta, and Ti. In summary, the dark luminescing domains of cassiterite grains from the Gejiu tin polymetallic deposit are mainly related with the concentrations of U and W. Comparatively, the regular oscillatory zoning pattern mainly correlates with the distribution of Nb, Ta, and Ti.

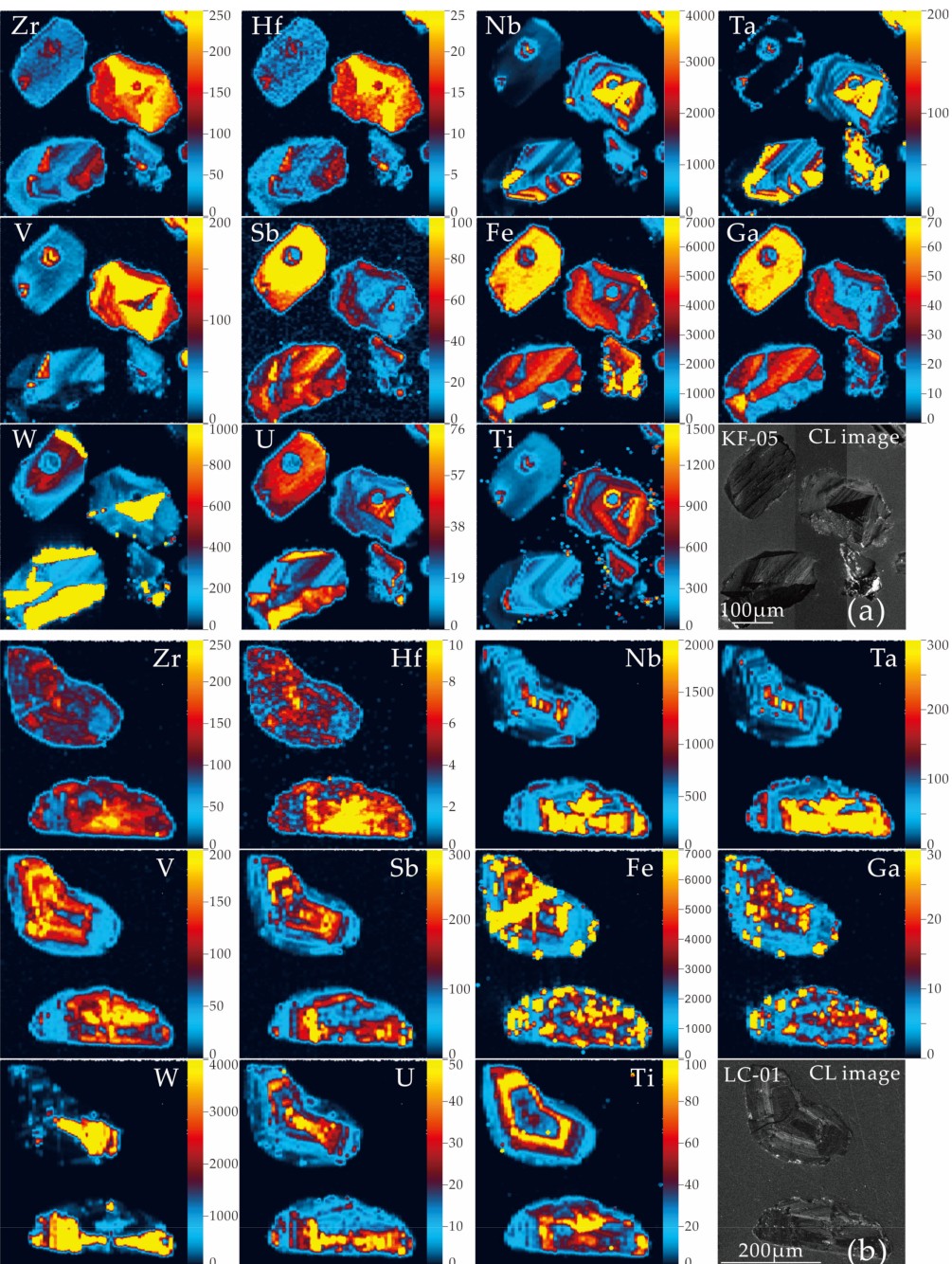

**Figure 6.** Coupled cathodoluminescence (CL) images and LA–ICP–MS multiple elemental mapping results of cassiterite from tin-granite (**a**), massive sulfide (**b**).

## 6. Discussion

### 6.1. Chronology of the Gejiu Tin Polymetallic Deposit

There is a wide range of views on ages of mineralization in the Gejiu tin polymetallic deposit. Qin et al. [20] used K–Ar and $^{40}$Ar–$^{39}$Ar dating on quartz/cassiterite and Pb–Pb dating on sulfide to argue for a broad mineralization age from 83.23 ± 2.07 to ~240 Ma. Yang et al. [21] performed Re–Os dating of molybdenite from the Kafang skarn Cu–Sn deposit, one of the four ore fields in the Gejiu tin polymetallic deposit, and obtained an isochron age of 83.4 ± 2.1 Ma. The $^{40}$Ar–$^{39}$Ar dating of muscovite from the muscovite–tourmaline–quartz vein ores in Laochang yielded a plateau age of 82.7 ± 0.7 Ma, which was considered as the mineralization age of the tin polymetallic deposit [45]. Li et al. [46] interpreted a cassiterite $^{40}$Ar–$^{39}$Ar isochron age of 206.8 ± 3.2 Ma and a $^{40}$Ar–$^{39}$Ar plateau age of 202.2 ± 2.4 Ma from the Lutangba orebody as the tin mineralization ages. Zhao et al. [10] collected two types of cassiterite from the Xi'ao Cu–Sn polymetallic deposit and acquired cassiterite U–Pb ages of 83.3 ± 2.1 and 84.9 ± 1.7 Ma, which are similar to the mineralization ages of the Gaosong Sn–Cu deposit (cassiterite U–Pb age of 83.5 ± 2.1–85.1 ± 1.0 Ma, after Guo et al. [8]). Of the various ages above derived from different methods, we prefer cassiterite U–Pb ages, which can be used directly to constrain the tin mineralization event [1,3,11,13,14,17,47] as the mineralization timing. The result of cassiterite U–Pb dating in the present study shows that Sn mineralization timing of the Gejiu tin polymetallic deposit falls within a narrow range of 83.4 ± 0.64 to 87.0 ± 1.6 Ma, in agreement with previously published cassiterite U–Pb ages of 76.4 ± 1.7 to 85.1 ± 1.0 Ma within their respective analytical uncertainties [3,8,10,22]. This indicates that tin mineralization occurred in the Late Cretaceous period within a relatively short time, suggesting that hydrothermal tin mineralization duration was short.

We note that no Triassic cassiterite U–Pb age has been reported up until now. This is contrary to the model proposed by Qin et al. [20,48], who argued that the stratiform-like tin mineralization of Triassic submarine exhalative sedimentary (SEDEX) origin had been modified by Cretaceous magmatic activity [23,34,46,49,50]. Given the age data presented here, this model may not be reasonable for the genesis of the Gejiu tin polymetallic deposit. Instead, our cassiterite U–Pb ages coincide with the mica $^{40}$Ar–$^{39}$Ar plateau ages (85.6 ± 0.7 Ma for the stratiform-like weathered ore and 84.3 ± 0.6 Ma for the skarn ore [28]) and zircon U–Pb ages of 77.4 ± 2.5 ~ 85.8 ± 0.6 Ma for the granitic pluton [30–33,51]. These ages suggest there is a close temporal and genetic relationship between Late Cretaceous granitic magmatism and tin mineralization in the Gejiu district. This viewpoint is consistent with the magmatic–hydrothermal model proposed by others [8,10,22,24,25,28,30–33,35,52,53]. The chemical composition of hydrothermal cassiterite can be a sensitive indicator for the mineralization type [54–56]. Enrichment of W in cassiterite from granite-related deposits and the limited Fe content in cassiterites from VMS/SEDEX deposits [55] are two key discriminating factors for determining genesis [8,55]. Almost all cassiterites in this study are characterized by high W and Fe contents and plot in the granite-related tin deposit area (Figure 7a), further suggesting that Gejiu tin polymetallic deposit is of magmatic–hydrothermal origin. This genetic model is supported by H–O stable isotopic composition of cassiterites ($\delta^{18}D_{H2O-SMOW}$ = −79–−153‰, $\delta^{18}O_{H2O-SMOW}$ = 7.16–8.25‰) coinciding with those of fluids derived from magmatic hydrothermal systems in available data [10,12,25,57].

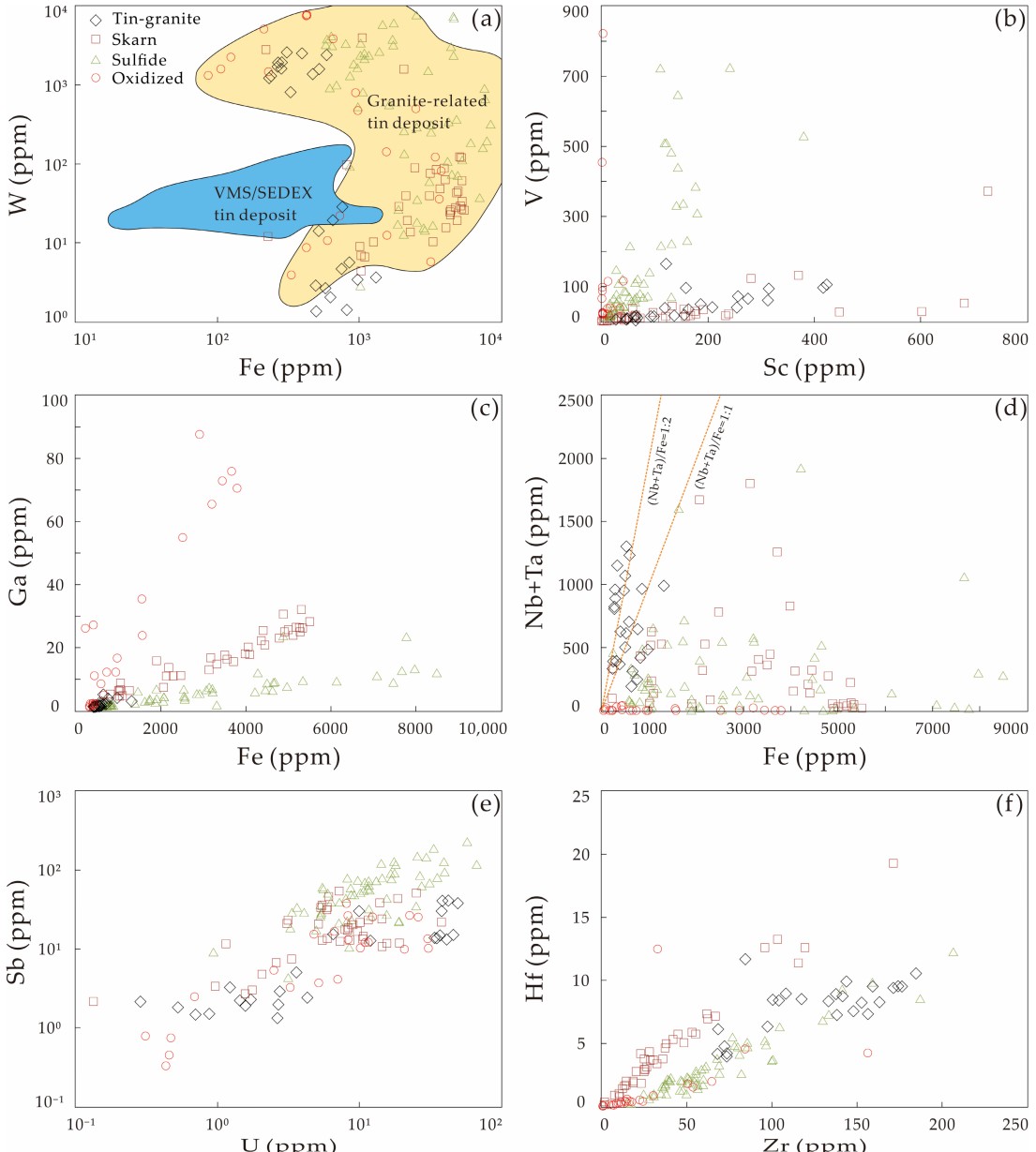

**Figure 7.** Correlation binary plots of selected trace elements in cassiterite samples from the Gejiu tin polymetallic deposit. (**a**) The plot of W vs. Fe displays that the tin mineralization is related to granitic magmatism; (**b**,**c**) The plots of V vs. Sc and Ga vs. Fe display a clear positive correlation; (**d**) The plot of Nb+Ta vs. Fe displays cassiterite from tin-granite have a (Nb + Ta)/Fe ratio near 1; (**e**,**f**) The plots of Sb vs. U and Hf vs. Zr display a positive correlation.

### 6.2. The Elemental Substitution Mechanism and Precipitation Environment of Cassiterites: Insights from Multiple Element Mapping and Trace Element Variations

Systematic variations in the chemical composition of cassiterite and substitution mechanisms are interpreted to be related to ore-forming environments and physicochemical conditions [5,54,56,58]. Considering the charge and radius of ions, cations such as $Fe^{3+}$, $Ga^{3+}$, $Al^{3+}$, $Sc^{3+}$, $Sb^{3+}$, $W^{4+}$, $U^{4+}$, $Zr^{4+}$, $Hf^{4+}$, $Ti^{4+}$, $V^{5+}$, $Nb^{5+}$, and $Ta^{5+}$ are considered compatible in cassiterite [59,60]. This is evidenced by high concentrations of these elements in cassiterite presented in previous studies [3,5,54,56,61]. Most of these compatible elements in the cassiterite samples examined here are in appreciable quantities and have significant compositional differences between different ore-type samples. These variations can provide indicative information on elemental substitution mechanisms in cassiterites.

Previously, elevated total Fe, Nb, Ta, Ti, Sc, and Mn contents were considered to play an important role in controlling color variation revealed by CL images of cassiterites [3,7,8,62]. Elemental mapping of cassiterite grains presented here indicates that the dark luminescing domains in CL are mainly related with elevated W and U as described above, whereas high concentrations of Ti and Mn are not necessary. Elemental maps further revealed the regular oscillatory zoning patterns mainly correlate with the distribution of Nb, Ta, and Ti (Figure 6a,b).

Elemental binary diagrams show a clear positive correlation between Sc and V (Figure 7b) and between Fe and Ga (Figure 7c) in the cassiterite samples analyzed here. Because Sc only has a single valence state (+3) under geological conditions [3], the positive Sc–V correlation indicates that V in the $5^+$ valence state incorporates in the cassiterite lattice, allowing a charge-balanced coupled substitution of $Sc^{3+} + V^{5+} \leftrightarrow 2\,(Sn, Ti)^{4+}$. Similarly, the significant correlation between Fe and Ga (Figure 7c) could be interpreted as the coupled substitution of $Fe^{3+} + Ga^{5+} \leftrightarrow 2\,(Sn, Ti)^{4+}$. However, the apparent higher content of Fe than that of Ga indicates that the incorporation of Fe might also follow another substitution: $Fe^{3+} + OH^- \leftrightarrow Sn^{4+} + O^{2-}$ or $Fe^{3+} + H^+ \leftrightarrow Sn^{4+}$ [54,63].

As valence state directly impacts the compatibility of an element in cassiterite [3,7], some redox sensitive elements such as W (+4 or +6), U (+4 or +6), Fe (+2 or +3), or Sb (+3 or +5) can be used as a powerful probe for evaluating the redox conditions of mineral precipitation processes. The distinct concentration variations of these elements in different domains of cassiterites are interpreted to reflect redox-driven chemical modification by reaction with hydrothermal fluid [3,7]. Combining the internal texture with the distribution and concentration of redox-sensitive elements in cassiterites, we can speculate on changes in redox and fluid mixing or reaction, which are considered as the most important drivers for the precipitation of tin ore [64]. Thus, this approach may be especially useful for understanding Sn mineralization systems [3]. There are two ideal coupled substitutions for incorporation of Nb and Ta in cassiterite: (1) $Fe^{2+} + 2(Nb, Ta)^{5+} \leftrightarrow 3\,(Sn, Ti)^{4+}$ ((Nb + Ta)/Fe $\approx$ 2) under a reduced condition; (2) $Fe^{3+} + (Nb, Ta)^{5+} \leftrightarrow 2\,(Sn, Ti)^{4+}$ ((Nb + Ta)/Fe $\approx$ 1) under an oxidized state [3,54,64–66], because isomorphous replacement of $Sn_3O_6$ by $(Fe,Mn)(Nb,Ta)_2O_6$ is typical of cassiterite in hydrothermal Sn deposits [54,56]. The incorporation of Nb and Ta depend greatly on the Fe valence, and consequently on the redox condition in the system. We can therefore define which substitution occurred in the cassiterite precipitation process according to the ratio of mol (Nb + Ta)/Fe. Only tin-granite cassiterite reveals (Nb + Ta)/Fe near 1 (Figure 7d), suggesting that tin-granite cassiterite was formed under an oxidized state with substitution (2) above. For magmatic–hydrothermal cassiterites from skarn, massive sulfide, and oxidized-type ores, there could be a more complex substitution mechanism, such as substitutions of $Fe^{3+} + OH^- \leftrightarrow Sn^{4+} + O^{2-}$ and $Fe^{3+} + H^+ \leftrightarrow Sn^{4+}$ (oxidizing environment) as proposed by previous studies [54,63,66,67]. However, under oxidized conditions, U exists in 5+ or 6+ state and is therefore incompatible in cassiterite, but is fluid mobile and transported out of the cassiterite [3,4,68]. The relatively elevated U and Sb contents and positive correlation (Figure 7e) suggest that these cassiterites were precipitated under a reduced environment, where $Sb^{3+}$ and $U^{4+}$ are more compatible than $Sb^{5+}$ and $U^{5+}$ (or $U^{6+}$). Zr and Hf have similar geochemical behavior, so they usually are constant in Zr/Hf ratio of 35–40 [69]. However, element pair decoupling can occur in hydrothermal environments and in highly differentiated igneous rocks [70–73]. Processes such as metasomatism, hydrothermal alteration, or preferential mobilization of Zr in B-rich and F-rich fluids have been proposed to cause fractionation of Zr over Hf [3,65]. Cassiterite samples studied here show a wide variation of Zr/Hf ratio (5–78, see Figure 7f), agreeing with the high activity of B and F during the Sn mineralization in Gejiu. This is also proved by the abundant tourmaline ($NaR_3Al_6Si_6O_{18}BO_{33}(OH,F)_4$) and fluorite ($CaF_2$) in ores reported in previous literature [3,8,10,22].

## 7. Conclusions

(1) Cassiterite U–Pb ages at around 85 Ma coincides with the crystallization ages of Late Cretaceous granitoids in the Gejiu district, suggesting that there is a temporal and spatial genetic relationship between Late Cretaceous granitic magmatism and tin mineralization.

(2) The distribution of Nb, Ta, and Ti in cassiterite grains correlate with regular oscillatory zoning patterns displayed by CL images. Comparatively, relatively high Sb, W, and U concentrations correlate with the dark luminescing domains in cassiterite grains.

(3) The variation relevance of redox sensitive elements such as W, U, Fe, and Sb suggest that tin-granite cassiterite was formed under an oxidized state whereas skarn, massive sulfide, and oxidized cassiterites were precipitated in a reduced environment.

**Author Contributions:** Data curation, S.T.; formal analysis, X.H.; funding acquisition, S.T. and X.H.; investigation, X.H., C.B., Y.L. and Z.L.; resources, S.T.; software, X.H.; supervision, S.T.; writing—original draft, X.H.; writing—review and editing, X.H. and N.L. All authors have read and agreed to the published version of the manuscript.

**Funding:** This study was supported by the National Natural Science Foundation of China (Grant No. 41702084, 41872089), Geology Discipline Construction Project of Yunnan University (Grant No. C176210227), and Yunnan Fundamental Research Projects (Grant No. 202101AT070011, 202101AW070012).

**Data Availability Statement:** The data presented in this study are available in this article.

**Conflicts of Interest:** The authors declare no conflict of interest.

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
