# Peer review of "LA–ICP–MS U–Pb Dating, Elemental Mapping and In Situ Trace Element Analyses of Cassiterites from the Gejiu Tin Polymetallic Deposit, SW China: Constraints on the Timing of Mineralization and Precipitation Environment"

_minerals, doi:10.3390/min12030313_

Round 1

Reviewer 1 Report

I suggest passing tables 1 and 2 to supporting information. And increase the graphic quality of figures 5 and 7.

Author Response

I suggest passing tables 1 and 2 to supporting information. And increase the graphic quality of figures 5 and 7.

Response: thank you for your good suggestions, low quality of figures may be caused by the submitting system, we have supplied higher resolution figure files to the editor office in this version.

Reviewer 2 Report

In the manuscript ID: minerals-1598513, the authors studied the dating and geochemistry of cassiterites from the Gejiu Tin deposit in the SW of China. This paper is interesting because it provides new insights into the origin of this deposit. The authors present quality data and content suitable for publication in Minerals. However, the manuscript requires some modifications before being published.

Most comments below are not criticisms to be addressed absolutely, but mere suggestions for improvement, and I trust the authors to know better than I which ones are valuable to follow.

COMMENTS

General

A valuable paper. The paper is competently written and easy to read and has been performed using appropriate techniques to arrive at the conclusions provided by the authors. The writing is clear, and the structure is satisfying, although syntactic and minor grammatical errors require a thorough review by the authors.

Using LA-ICP-MS combined with CL, the authors address the age and formation conditions of the cassiterites. However, the paper could be improved by including some general data on the mineralogy and grades of the deposit. A brief petrographic description of the samples used in the study is also missing. 

They synthesise information obtained as tables, pictures, and diagrams. However, some of these figures and charts should be improved, as indicated below. The authors include almost all the data obtained in the form of tables in the text, which are perhaps a little distracting for the reader. These tables would probably be better included in an annexe. The patterns extracted from the data are carefully handled. 

Main Comments

My main comments on the paper are: 

1) The paper is well structured.

2) Figures are sufficient and perfectly support the authors' arguments. They support the explanation well, except for some discussed later. 

3) The methodology for obtaining the data is adequate for the work in question.

4) Although it is my opinion, the authors can improve the writing of the conclusions by avoiding stating each finding as a point not connected to the previous one. I believe that with a bit of effort, the findings would be much more attractive.

Although all comments are included in the attached document, here I highlight some that could be taken into consideration:

  • Line 34: Specify the style of mineralisation.
  • Line 48: Reference 16 is missing; therefore, from here, you should correct the numbering of all references.
  • Line 55: Consider using a more geological term, such as minerals or elements.
  • Line 56: You mean that there are some studies on fluid inclusions. Please indicate them.
  • Line 58: Figure 1a is not clearly visible. If you think it is essential, you should make it larger to read its contents.
  • Line 158: Figure 3 should be improved. The lighting in the first three photographs is very poor, despite being taken in an enclosed space. Try to uniformly illuminate the specimens, or at least, do not place the light source at the back of the sample. The last photo, being from the field, has an important limitation to obtaining good illumination. Also, if you use a white background for the rock samples, please make sure that nothing but your designated background is visible. This will considerably improve the quality of the photographs.
  • Line 222: Why do you use 1< in W and Nb and not 0, as in the rest of the elements? Is this difference in rank justifiable?
  • Line 231: In table 2, you combine several decimal places for the same element. Please, it should be uniform. Moreover, in the composition of Al2O3 (and some other elements), you use up to four decimal places. Is it necessary, is there analytical certainty in these values, and is the level of decimal places relevant for discussing the composition of cassiterites?
  • Line 279: This sentence raises some doubts. You should be more precise and more direct. Perhaps something like "No Triassic cassiterite U-Pb age has been reported in the Gejiu deposit". As you put it there is a possibility that Triassic cassiterites exist and therefore your argumentation is a bit weaker.
  • Line 281: You should indicate "exhalative sedimentary", as exhalative submarine can also be VMS deposits.
  • Line 328: There is a sharpness problem with this figure. It was probably a problem with the layout by the editorial and not of the authors. There are trends in these figures that should be discussed in the text because in figures 7b, c and f, there are strong trends that would require explanation. And this explanation would undoubtedly enrich the discussion and conclusions.

All other comments are included in the two attached documents. I am sorry, but I had a computer problem, so I am sending you two documents.

References 

The bibliography has been well used, including the most relevant papers in the field. The authors follow the Minerals standards for reference, and I have only detected minor errors in the attached file.

Typos and nitpicking

Typographical errors and nitpicking are included in the attachment. I also include some suggestions about the text.

Author Response

Response: Thank you very much for your good suggestions, we have revised the text according your annotations.

Reviewer 3 Report

See comments attached in edited PDF. The data and interpretation of results are significant. The graphics are blurry and need to be higher resolution. The correlation between redox state and geochemical ratio interpretation can be improved by provided graphical representation of ratios and reduced/oxidized zones for each type to better demonstrate the differences and similarities.

Author Response

See comments attached in edited PDF. The data and interpretation of results are significant. The graphics are blurry and need to be higher resolution. The correlation between redox state and geochemical ratio interpretation can be improved by provided graphical representation of ratios and reduced/oxidized zones for each type to better demonstrate the differences and similarities.

Response: thank you for your good suggestions, low quality of figures may be caused by the submitting system, we have supplied higher resolution figure files to the editor office, revised spelling mistakes and other text according to your comments annotated in PDF file in this version.

Reviewer 4 Report

It is more convenient for the readers to be able to download a data file (Table 2. Trace element composition of tin-granite, skarn and massive sulfide types of cassiterite 231 samples analyzed by LA-ICP-MS single spot-analysis).

Author Response

It is more convenient for the readers to be able to download a data file (Table 2. Trace element composition of tin-granite, skarn and massive sulfide types of cassiterite 231 samples analyzed by LA-ICP-MS single spot-analysis).

Response: thank you for your good suggestions, we have supplied table files to editor office in this version.

Round 2

Reviewer 1 Report

No comments

Author Response

Response: thank you very much

Reviewer 2 Report

Dear authors: 

Thank you very much for the effort made and for your reply. This reply has clarified your approach, and it is very well justified and argued. I consider that the work carried out meets the quality parameters required by Minerals for publication.

I only suggest that you improve the tables so that the final result is sufficient for this journal. Please note that each column has the same decimal places and that no part of the table appears without a header.

In the attached document, I include some notes if you can consider any of the suggestions included.

Congratulations on your work, and thank you for viewing some of my suggestions.

Author Response

Response: Thank you very much for your good suggestions. we have revised the text according to your annotations again. The format of tables is typeset by the editor, thus we will give your suggestion to him/her. In addition, we think how many decimal of a valid number should depend on the accuracy and error of the instrument, uniform all numbers to 2 or 3 decimal may be rough and simple. Thus, using significant figures is correct. In the table 2, three significant figure is suitable and scientific. Thank you again.

Reviewer 3 Report

Incorporation of comments and suggestions has improved the manuscript, and improved clarity of figures is much appreciated.

Author Response

Response: thank you very much

Reviewer 4 Report

The article contains a lot of new data that will be useful. Accept in present form.

Author Response

Response: thank you very much